# Sparse communication via mixed distributions

**António Farinhas [1], Wilker Aziz [2], Vlad Niculae [3], André F. T. Martins [1,4]**
[1]Instituto de Telecomunicações, Instituto Superior Técnico (Lisbon ELLIS Unit),
[2]ILLC, University of Amsterdam,  [3]IvI, University of Amsterdam,  [4]Unbabel
{antonio.farinhas,andre.t.martins}@tecnico.ulisboa.pt, {w.aziz,v.niculae}@uva.nl

## Abstract

Neural networks and other machine learning models compute continuous representations, while humans communicate mostly through discrete symbols. Reconciling these two forms of communication is desirable for generating human-readable interpretations or learning discrete latent variable models, while maintaining end-to-end differentiability. Some existing approaches (such as the Gumbel-Softmax transformation) build continuous relaxations that are discrete approximations in the zero-temperature limit, while others (such as sparsemax transformations and the Hard Concrete distribution) produce discrete/continuous hybrids. In this paper, we build rigorous theoretical foundations for these hybrids, which we call "mixed random variables." Our starting point is a new "direct sum" base measure defined on the face lattice of the probability simplex. From this measure, we introduce new entropy and Kullback-Leibler divergence functions that subsume the discrete and differential cases and have interpretations in terms of code optimality. Our framework suggests two strategies for representing and sampling mixed random variables, an extrinsic ("sample-and-project") and an intrinsic one (based on face stratification). We experiment with both approaches on an emergent communication benchmark and on modeling MNIST and Fashion-MNIST data with variational auto-encoders with mixed latent variables. Our code is publicly available.

## 1 Introduction

Historically, discrete and continuous domains have been considered separately in machine learning, information theory, and engineering applications: random variables (r.v.) and information sources are chosen to be either discrete or continuous, but not *both* (Shannon, 1948). In signal processing, one needs to opt between discrete (digital) and continuous (analog) communication, whereas analog signals can be converted into digital ones by means of sampling and quantization.

Discrete latent variable models are appealing to facilitate learning with less supervision, to leverage prior knowledge, and to build more compact and interpretable models. However, training such models is challenging due to the need to evaluate a large or combinatorial expectation. Existing strategies include the score function estimator (Williams, 1992; Mnih & Gregor, 2014), pathwise gradients (Kingma & Welling, 2014) combined with a continuous relaxation of the latent variables (such as the Concrete distribution, Maddison et al. (2017); Jang et al. (2017)), and sparse parametrizations (Correia et al., 2020). Pathwise gradients, in particular, require continuous approximations of quantities that are inherently discrete, sometimes requiring proxy gradients (Jang et al., 2017; Maddison et al., 2017), sometimes giving the r.v. different treatment in different terms of the same objective (Jang et al., 2017), sometimes creating a discrete-continuous hybrid (Louizos et al., 2018).

Since discrete variables and their continuous relaxations are so prevalent, they deserve a rigorous mathematical study. Throughout, we will use the name **mixed variable** to denote a hybrid variable that takes on both discrete and continuous outcomes. This work takes a first step into a rigorous study of mixed variables and their properties. We will call communication through mixed variables **sparse communication**: its goal is to retain the advantages of differentiable computation but still be able to represent and approximate discrete symbols. Our main contributions are:

- We provide a **direct sum measure** as an alternative to the Lebesgue and counting measures used for continuous and discrete variables, respectively (Halmos, 2013). The direct sum measure hinges

Table 1: Discrete, continuous, and mixed distributions considered in this work, all involving the probability simplex $\triangle_{K-1}$. For each distribution we indicate if it assigns probability mass to all faces of the simplex or only some, if it is multivariate ($K \geq 2$), and, for mixed distributions, if it is characterized extrinsically (sample-and-project) or intrinsically (based on face stratification).

| Distribution | All faces? | Multivariate? | Intrinsic? |
|---|---|---|---|
| Categorical | Discrete ✗ | Yes ✓ | – |
| Dirichlet, Logistic-Normal, Concrete | Continuous ✗ | Yes ✓ | – |
| Hard Concrete, Rectified Gaussian | Mixed ✓ | No ✗ | Extrinsic ✗ |
| $K$-D Hard Concrete, Gaussian-Sparsemax (this paper) | Mixed ✓ | Yes ✓ | Extrinsic ✗ |
| Mixed Dirichlet (this paper) | Mixed ✓ | Yes ✓ | Intrinsic ✓ |

on a **face lattice stratification** of polytopes, including the probability simplex, avoiding the need for Dirac densities when expressing densities with point masses in the boundary of the simplex (§3).

• We use the direct sum measure to formally define $K^{\text{th}}$-dimensional **mixed random variables**. We provide extrinsic ("sample-and-project") and intrinsic (based on face stratification) characterizations of these variables, leading to several new distributions: the K-D Hard Concrete, the Gaussian-Sparsemax, and the Mixed Dirichlet (summarized in Table 1). See Figure 1 for an illustration.

• We propose a new **direct sum entropy and Kullback-Leibler divergence**, which decompose as a sum of discrete and continuous (differential) entropies/divergences. We provide an interpretation in terms of optimal code length, and we derive an expression for the maximum entropy (§4).

• We illustrate the usefulness of our framework by learning **mixed latent variable models** in an emergent communication task and with VAEs to model Fashion-MNIST and MNIST data (§5).

## 2 BACKGROUND

We assume throughout an alphabet with $K \geq 2$ symbols, denoted $[K] = \{1, \ldots, K\}$. Symbols can be encoded as one-hot vectors $\boldsymbol{e}_k$. $\mathbb{R}^K$ denotes the $K$-dimensional Euclidean space, $\mathbb{R}^K_{>0}$ its strictly positive orthant, and $\triangle_{K-1} \subseteq \mathbb{R}^K$ the **probability simplex**, $\triangle_{K-1} := \{\boldsymbol{y} \in \mathbb{R}^K \mid \boldsymbol{y} \geq \boldsymbol{0}, \ \boldsymbol{1}^\top \boldsymbol{y} = 1\}$, with vertices $\{\boldsymbol{e}_1, \ldots, \boldsymbol{e}_K\}$. Each $\boldsymbol{y} \in \triangle_{K-1}$ can be seen as a vector of probabilities for the $K$ symbols, parametrizing a categorical distribution over $[K]$. The **support** of $\boldsymbol{y} \in \triangle_{K-1}$ is the set of nonzero-probability symbols $\mathrm{supp}(\boldsymbol{y}) := \{k \in [K] \mid y_k > 0\}$. The set of full-support categoricals corresponds to the **relative interior** of the simplex, $\mathrm{ri}(\triangle_{K-1}) := \{\boldsymbol{y} \in \triangle_{K-1} \mid \mathrm{supp}(\boldsymbol{y}) = [K]\}$.

### 2.1 TRANSFORMATIONS FROM $\mathbb{R}^K$ TO $\triangle_{K-1}$

In many situations, there is a need to convert a vector of real numbers $\boldsymbol{z} \in \mathbb{R}^K$ (scores for the several symbols, often called *logits*) into a probability vector $\boldsymbol{y} \in \triangle_{K-1}$. The most common choice is the **softmax** transformation (Bridle, 1990): $\boldsymbol{y} = \mathrm{softmax}(\boldsymbol{z}) \propto \exp(\boldsymbol{z})$. Since the exponential function is strictly positive, softmax reaches only the relative interior $\mathrm{ri}(\triangle_{K-1})$, that is, it never returns a sparse probability vector. To encourage more peaked distributions (but never sparse) it is common to use a **temperature** parameter $\beta > 0$, by defining $\mathrm{softmax}_\beta(\boldsymbol{z}) := \mathrm{softmax}(\beta^{-1}\boldsymbol{z})$. The limit case $\beta \to 0_+$ corresponds to the indicator vector for the **argmax**, which returns a one-hot distribution indicating the symbol with the largest score. While the softmax transformation is differentiable (hence permitting end-to-end training with the gradient backpropagation algorithm), the argmax function has zero gradients almost everywhere. With small temperatures, numerical issues are common.

A direct sparse probability mapping is **sparsemax** (Martins & Astudillo, 2016), the Euclidean projection onto the simplex: $\mathrm{sparsemax}(\boldsymbol{z}) := \arg\min_{\boldsymbol{y} \in \triangle_{K-1}} \|\boldsymbol{y} - \boldsymbol{z}\|$. Unlike softmax, sparsemax reaches the *full* simplex $\triangle_{K-1}$, including the boundary, often returning a sparse vector $\boldsymbol{y}$, without sacrificing differentiability almost everywhere. With $K = 2$ and parametrizing $\boldsymbol{z} = (z, 1 - z)$, sparsemax becomes a "hard sigmoid," $\left[\mathrm{sparsemax}\big((z, 1 - z)\big)\right]_1 = \max\{0, \min\{1, z\}\}$. We will come back to this point in §3.3. Other sparse transformations include $\alpha$-entmax (Peters et al., 2019; Blondel et al., 2020), top-$k$ softmax (Fan et al., 2018; Radford et al., 2019), and others (Laha et al., 2018; Sensoy et al., 2018; Kong et al., 2020; Itkina et al., 2020).

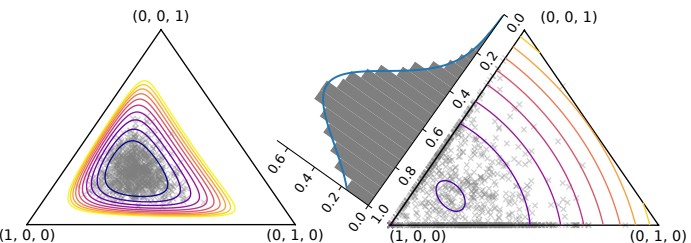

Figure 1: Multivariate distributions over $\triangle_{K-1}$. Standard distributions, like the Logistic-Normal (left), assign zero probability to all faces but $\mathrm{ri}(\triangle_{K-1})$. Our **mixed distributions** support assigning probability to the *full* simplex, including its boundary: the Gaussian-Sparsemax (right) induces a distribution over the 1-dimensional edges (shown as a histogram), and assigns $\Pr\{(1,0,0)\} = .022$.

## 2.2 DENSITIES OVER THE SIMPLEX

Let us now switch from deterministic to *stochastic* maps. Denote by $Y$ a r.v. taking on values in the simplex $\triangle_{K-1}$ with probability density function $p_Y(\boldsymbol{y})$.

The density of a **Dirichlet** r.v. $Y \sim \mathrm{Dir}(\boldsymbol{\alpha})$, with $\boldsymbol{\alpha} \in \mathbb{R}^K_{>0}$ is $p_Y(\boldsymbol{y}; \boldsymbol{\alpha}) \propto \prod_{k=1}^K y_k^{\alpha_k - 1}$. Sampling from a Dirichlet produces a point in $\mathrm{ri}(\triangle_{K-1})$, and, although a Dirichlet can assign high density to $\boldsymbol{y}$ close to the boundary of the simplex when $\boldsymbol{\alpha} < \mathbf{1}$, a Dirichlet sample can *never* be sparse.

A **Logistic-Normal** r.v. (Atchison & Shen, 1980), also known as Gaussian-Softmax by analogy to other distributions to be presented, is given by the softmax-projection of a multivariate Gaussian r.v. with mean $\boldsymbol{z}$ and covariance $\Sigma$: $Y = \mathrm{softmax}(\boldsymbol{z} + \Sigma^{\frac{1}{2}} N)$ with $N_k \sim \mathcal{N}(0,1)$. Since the softmax is strictly positive, the Logistic-Normal places no probability mass to points in the boundary of $\triangle_{K-1}$.

A **Concrete** (Maddison et al., 2017), or Gumbel-Softmax (Jang et al., 2017), r.v. is given by the softmax-projection of $K$ independent Gumbel r.vs., each with mean $z_k$: $Y = \mathrm{softmax}_\beta(\boldsymbol{z} + G)$ with $G_k \sim \mathrm{Gumbel}(0,1)$. Like in the previous cases, a Concrete draw is a point in $\mathrm{ri}(\triangle_{K-1})$. When the temperature $\beta$ approaches zero, the softmax approaches the indicator for argmax and $Y$ becomes closer to a categorical r.v. (Luce, 1959; Papandreou & Yuille, 2011). Thus, a Concrete r.v. can be seen as a *continuous* relaxation of a categorical.

## 2.3 TRUNCATED UNIVARIATE DENSITIES

**Binary Hard Concrete.** For $K = 2$, a point in the simplex can be represented as $\boldsymbol{y} = (y, 1-y)$ and the simplex is isomorphic to the unit interval, $\triangle_1 \simeq [0,1]$. For this binary case, Louizos et al. (2018) proposed a *Hard Concrete distribution* which stretches the Concrete and applies a hard sigmoid transformation (which equals the sparsemax with $K = 2$, per §2.1) as a way of placing point masses at 0 and 1. These "stretch-and-rectify" techniques enable assigning probability mass to the boundary of $\triangle_1$ and are similar in spirit to the spike-and-slab feature selection method (Mitchell & Beauchamp, 1988; Ishwaran et al., 2005) and for sparse codes in variational auto-encoders (Rolfe, 2017; Vahdat et al., 2018). We propose in §3.3 a more general extension to $K \geq 2$.

**Rectified Gaussian.** Rectification can be applied to other continuous distributions. A simple choice is the Gaussian distribution, to which one-sided (Hinton & Ghahramani, 1997) and two-sided rectifications (Palmer et al., 2017) have been proposed. Two-sided rectification yields a mixed r.v. in $[0,1]$. Writing $\boldsymbol{y} = (y, 1-y)$ and $\boldsymbol{z} = (z, 1-z)$, this distribution has the following density:

$$p_Y(y) = \mathcal{N}(y; z, \sigma^2) + \frac{1 - \mathrm{erf}(z/(\sqrt{2}\sigma))}{2}\delta_0(y) + \frac{1 + \mathrm{erf}((z-1)/(\sqrt{2}\sigma))}{2}\delta_1(y), \quad (1)$$

where $\delta_s(y)$ is a Dirac delta density. Extending such distributions to the multivariate case is non-trivial. For $K > 2$, a density expression with Diracs would be cumbersome, since it would require a combinatorial number of Diracs of several "orders," depending on whether they are placed at a vertex, edge, face, etc. Another annoyance is that Dirac deltas have $-\infty$ differential entropy, which prevents information-theoretic treatment. The next section shows how we can obtain densities that assign mass to the full simplex while avoiding Diracs, by making use of the face lattice and a new base measure.

## 3 FACE STRATIFICATION AND MIXED RANDOM VARIABLES

### 3.1 THE FACE LATTICE

Let $\mathcal{P}$ be a **convex polytope** whose vertices are bit vectors (*i.e.*, elements of $\{0,1\}^K$). Examples are the probability simplex $\triangle_{K-1}$, the hypercube $[0,1]^K$, and marginal polytopes of structured variables (Wainwright & Jordan, 2008). The combinatorial structure of a polytope is determined by its **face lattice** (Ziegler, 1995, §2.2), which we now describe. A **face** of $\mathcal{P}$ is any intersection of $\mathcal{P}$ with a closed halfspace such that none of the interior points of $\mathcal{P}$ lie on the boundary of the halfspace; we denote by $\mathcal{F}(\mathcal{P})$ the set of *all faces* of $\mathcal{P}$ and by $\bar{\mathcal{F}}(\mathcal{P}) := \mathcal{F}(\mathcal{P}) \setminus \{\varnothing\}$ the set of proper faces. We denote by $\dim(f)$ the **dimension** of a face $f \in \bar{\mathcal{F}}(\mathcal{P})$. Thus, the vertices of $\mathcal{P}$ are 0-dimensional faces, and $\mathcal{P}$ itself is a face of the same dimension as $\mathcal{P}$, called the "maximal face". Any other face of $\mathcal{P}$ can be regarded as a lower-dimensional polytope. The set $\mathcal{F}(\mathcal{P})$ has a partial order induced by set inclusion, that is, it is a partially ordered set (poset), and more specifically a *lattice*. The full polytope $\mathcal{P}$ can be decomposed uniquely as the **disjoint union** of the relative interior of its faces, which we call **face stratification**: $\mathcal{P} = \bigsqcup_{f \in \bar{\mathcal{F}}(\mathcal{P})} \mathrm{ri}(f)$. For example, the simplex $\triangle_2$ is composed of its face $\mathrm{ri}(\triangle_2)$ (*i.e.*, excluding the boundary), three edges (excluding the vertices in the corners), and three vertices (the corners). This is represented schematically in Figure 2. Likewise, the square $[0,1]^2$ is composed of its maximal face $(0,1)^2$, four edges (excluding the corners) and four vertices (the corners). The partition above implies that any subset $A \subseteq \mathcal{P}$ can be represented as a tuple $A = (A_f)_{f \in \bar{\mathcal{F}}(\mathcal{P})}$, where $A_f = A \cap \mathrm{ri}(f)$; and the sets $A_f$ are all disjoint.

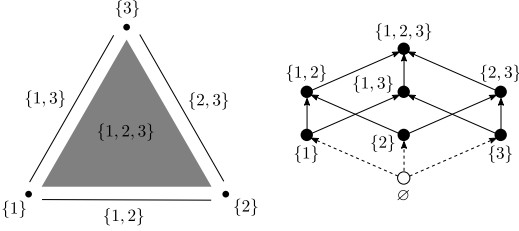

Figure 2: Left: Decomposition of a simplex as the disjoint union of the relative interior of its faces. Right: Hasse diagram of the face lattice – a DAG where points represent faces, and a directed path from a point to another represents face inclusion.

**Simplex and hypercube.** If $\mathcal{P}$ is the simplex $\triangle_{K-1}$, each face corresponds to an index set $\mathcal{I} \subseteq [K]$, *i.e.*, it can be expressed as $f_{\mathcal{I}} = \{\boldsymbol{p} \in \triangle_{K-1} \mid \mathrm{supp}(\boldsymbol{p}) \subseteq \mathcal{I}\}$, with dimension $\dim(f_{\mathcal{I}}) = |\mathcal{I}| - 1$: the set of distributions assigning zero probability mass outside $\mathcal{I}$. The set $\bar{\mathcal{F}}(\triangle_{K-1})$ has $2^K - 1$ elements. Since $\triangle_1 \simeq [0,1]$, the *hypercube* $[0,1]^K$ can be regarded as a product of $K$ binary probability simplices. It has $3^K$ nonempty faces – for each dimension we choose between $\{0\}$, $\{1\}$, and $[0,1]$. We experiment with $\mathcal{P} \in \{\triangle_{K-1}, [0,1]^K\}$ in §5.

### 3.2 MIXED RANDOM VARIABLES

Categorical distributions assign probability only to the vertices of $\triangle_{K-1}$. In the opposite extreme, the densities listed in §2.2 assign probability mass to the maximal face only, that is, $\Pr\{\boldsymbol{y} \in f_{\mathcal{I}}\} = \int_{f_{\mathcal{I}}} p_Y(\boldsymbol{y}) = 0$ for any $\mathcal{I} \neq [K]$. Any proper density (without Diracs) has this limitation, since non-maximal faces have zero Lebesgue measure in $\mathbb{R}^{K-1}$. While for $K = 2$ it is possible to circumvent this by defining densities that contain Dirac functions (as in §2.3), this becomes cumbersome for $K > 2$. Fortunately, there is a more elegant construction that does not require generalized functions. The key is to replace the Lebesgue measure by a measure inspired by face stratification.

**Definition 1** (Direct sum measure). *The direct sum measure on a polytope $\mathcal{P}$ is*

$$\mu^{\oplus}(A) = \sum_{f \in \bar{\mathcal{F}}(\mathcal{P})} \mu_f(A \cap \mathrm{ri}(f)), \tag{2}$$

*where $\mu_f$ is the $\dim(f)$-dimensional Lebesgue measure for $\dim(f) > 0$, and the counting measure for $\dim(f) = 0$.*

We show in App. A that $\mu^{\oplus}$ is a valid measure on $\mathcal{P}$ under the product $\sigma$-algebra of its faces. We can then define probability densities $p_Y^{\oplus}(\boldsymbol{y})$ w.r.t. this base measure and use them to compute probabilities of measurable subsets of $\mathcal{P}$. Such distributions can equivalently be defined as follows: *(i)* define a probability mass function $P_F(f)$ on $\bar{\mathcal{F}}(\mathcal{P})$, and *(ii)* for each face $f \in \bar{\mathcal{F}}(\mathcal{P})$, define a probability

density $p_{Y|F}(\boldsymbol{y} \mid f)$ over $\mathrm{ri}(f)$. Random variables with a distribution of this form have a discrete part and a continuous part, thus we call them **mixed random variables**. This is formalized as follows.

> **Definition 2** (Mixed random variable). *A mixed r.v. is a r.v. $Y$ over a polytope $\mathcal{P}$, including the boundary. Let $F$ be the corresponding discrete r.v. over $\bar{\mathcal{F}}(\mathcal{P})$, with $P_F(f) = \Pr\{\boldsymbol{y} \in \mathrm{ri}(f)\}$. Since the mapping $Y \to F$ is deterministic, we have $p_Y^{\oplus}(\boldsymbol{y}) = P_F(f)p_{Y|F}(\boldsymbol{y} \mid f)$ for $\boldsymbol{y} \in \mathrm{ri}(f)$. The probability of a set $A \subseteq \mathcal{P}$ is given by:*
>
> $$\Pr\{\boldsymbol{y} \in A\} = \int_A p_Y^{\oplus}(\boldsymbol{y})\mathrm{d}\mu^{\oplus} = \sum_{f \in \bar{\mathcal{F}}(\mathcal{P})} P_F(f) \int_{A \cap \mathrm{ri}(f)} p_{Y|F}(\boldsymbol{y} \mid f). \tag{3}$$

Equation (3) may be regarded as a manifestation of the law of total probability mixing discrete and continuous variables. Using (3), we can write expectations over a mixed r.v. as

$$\mathbb{E}_{p_Y^{\oplus}}[g(Y)] = \mathbb{E}_{P_F}\left[\mathbb{E}_{p_{Y|F}}[g(Y) \mid F = f]\right], \quad \text{where } g : \mathcal{P} \to \mathbb{R}. \tag{4}$$

Both discrete and continuous distributions are recovered with our definition: If $P_F(f) = 0$ for $\dim(f) > 0$, we have a categorical distribution, which only assigns probability to the 0-faces. In the other extreme, if $P_F(\mathcal{P}) = 1$, we have a continuous distribution confined to $\mathrm{ri}(\mathcal{P})$. That is, **mixed random variables include purely discrete and purely continuous r.vs. as particular cases.** To parametrize distributions of high-dimensional mixed r.vs., it is not efficient to consider all degrees of freedom suggested in Definition 2, since there can be exponentially many faces. Instead, we need to derive parametrizations that exploit the lattice structure of the faces. We next build upon this idea.

### 3.3 EXTRINSIC AND INTRINSIC CHARACTERIZATIONS

There are two possible characterizations of mixed random variables: an **extrinsic** one, where one starts with a distribution over the ambient space (*e.g.* $\mathbb{R}^K$) and then applies a deterministic, non-invertible, transformation that projects it to $\mathcal{P}$; and an **intrinsic** one, where one specifies a mixture of distributions directly over the faces of $\mathcal{P}$, by specifying $P_F$ and $p_{Y|F}$ for each $f \in \bar{\mathcal{F}}(\mathcal{P})$. We next provide constructions for both cases: We extend the Hard Concrete and Rectified Gaussian distributions reviewed in §2.3, which are instances of the extrinsic characterization, to $K \geq 2$; and we present a new Mixed Dirichlet distribution which is an instance of the intrinsic characterization.

**$K$-D Hard Concrete.** We define the $K$-D Hard Concrete as the following generative story:

$$Y' \sim \mathrm{Concrete}(\boldsymbol{z}, \beta), \quad Y = \mathrm{sparsemax}(\lambda Y'), \quad \text{with } \lambda \geq 1. \tag{5}$$

When $K = 2$, sparsemax becomes a hard sigmoid and we recover the binary Hard Concrete (§2.3). For $K \geq 2$ this is a projection of a "stretched" Concrete r.v. onto the simplex – the larger $\lambda$, the higher the tendency of this projection to hit a non-maximal face of the simplex and induce sparsity.

**Gaussian-Sparsemax.** A similar idea (but without any stretching required) can be used to obtain a sparsemax counterpart of the Logistic-Normal in §2.2, which we call "Gaussian-Sparsemax":

$$N \sim \mathcal{N}(\mathbf{0}, \mathbf{I}), \quad Y = \mathrm{sparsemax}(\boldsymbol{z} + \Sigma^{1/2} N). \tag{6}$$

Unlike the Logistic-Normal, the Gaussian-Sparsemax can assign nonzero probability mass to the boundary of the simplex. When $K = 2$, we recover the double-sided rectified Gaussian described in §2.3. In that case, using Dirac deltas, the density with respect to the Lebesgue measure in $\mathbb{R}$ has the form in (1). With $\theta_0 = \frac{1 - \mathrm{erf}(z/(\sqrt{2}\sigma))}{2}$, $\theta_1 = \frac{1 + \mathrm{erf}((z-1)/(\sqrt{2}\sigma))}{2}$ and $\theta_c = 1 - \theta_0 - \theta_1$, the same distribution can be expressed *intrinsically* via the density $p_Y^{\oplus}(y) = P_F(f)p_{Y|F}(\boldsymbol{y} \mid f)$ as

$$P_F(\{0\}) = \theta_0, \; P_F(\{1\}) = \theta_1, \; P_F([0,1]) = \theta_c, \; p_{Y|F}(y \mid F = [0,1]) = \mathcal{N}(y;z,\sigma^2)/\theta_c. \tag{7}$$

For $K > 2$, expressions for $P_F$ and $p_{Y|F}$ (*i.e.*, an intrinsic representation) are less direct; we express those distributions as a function of the orthant probability of multivariate Gaussians in App. B.

**Mixed Dirichlet.** We now propose an *intrinsic* mixed distribution over $\triangle_{K-1}$, the Mixed Dirichlet, whose generative story is as follows. First, a face $F = f_{\mathcal{I}}$ is sampled with probability

$$P_F(f_{\mathcal{I}}; \boldsymbol{w}) = \exp(\boldsymbol{w}^\top \boldsymbol{\phi}(f_{\mathcal{I}}) - \log Z(\boldsymbol{w})), \quad \phi_k(f_{\mathcal{I}}) = (-1)^{1-[k \in \mathcal{I}]}, \tag{8}$$

where $\boldsymbol{w} \in \mathbb{R}^K$ is the natural parameter (a.k.a. *log-potentials*), $\boldsymbol{\phi}(f_{\mathcal{I}}) \in \{-1, 1\}^K$ is the sufficient statistic, and $\log Z(\boldsymbol{w})$ is the log-normalizer. We then parametrize a Dirichlet distribution over the relative interior of $f_{\mathcal{I}}$, that is, $Y|F = f_{\mathcal{I}} \sim \mathrm{Dir}(\boldsymbol{\alpha}(\mathcal{I}))$, where $\boldsymbol{\alpha}(\mathcal{I}) \in \mathbb{R}^{|\mathcal{I}|}_{>0}$. For a compact parametrization, we have a single $K$-dimensional vector $\boldsymbol{\alpha}$ of concentration parameters, one parameter per vertex, and $\boldsymbol{\alpha}(\mathcal{I})$ gathers the coordinates of $\boldsymbol{\alpha}$ associated with the vertices in $f$. The normalizer of (8) can be evaluated in time $\mathcal{O}(K)$ via the forward algorithm (Baum & Eagon, 1967) on a directed acyclic graph (DAG) that encodes each non-empty corner $\mathcal{I} \subseteq [K]$ of the face lattice as a path. Similarly, we can draw independent samples by stochastic traversals through this DAG. The graph needed for this construct and the associated algorithms are detailed in App. C.

# 4 INFORMATION THEORY FOR MIXED RANDOM VARIABLES

Now that we have the theoretical pillars for mixed random variables, we proceed to defining information theoretic quantities for them: their entropy and Kullback-Leibler divergence.

**Direct sum entropy and KL divergence.** The entropy of a r.v. $X$ with respect to a measure $\mu$ is:

$$H^\mu(X) = -\int_{\mathcal{X}} p_X(x) \log p_X(x) \mathrm{d}\mu(x), \tag{9}$$

where $p_X(x)$ is a probability density satisfying $\int_{\mathcal{X}} p_X(x) \mathrm{d}\mu(x) = 1$. When $\mathcal{X}$ is finite and $\mu$ is the counting measure, the integral becomes a sum and we recover **Shannon's discrete entropy**, which is non-negative and upper bounded by $\log |\mathcal{X}|$. When $\mathcal{X} \subseteq \mathbb{R}^k$ is continuous and $\mu$ is the Lebesgue measure, we recover the **differential entropy**, which can be negative and, for compact $\mathcal{X}$, is upper bounded by the logarithm of the volume of $\mathcal{X}$. When relaxing a discrete r.v. to a continuous one in a variational model (*e.g.*, using the Concrete distribution), correct variational lower bounds require switching to differential entropy (Maddison et al., 2017) (although discrete entropy is sometimes used (Jang et al., 2017)). This is problematic since the differential entropy is not a limit case of the discrete entropy (Cover & Thomas, 2012). Our direct sum entropy, defined below, obviates this. The key idea is to plug in (9) the direct sum measure (2). Since $Y \rightarrow F$ is deterministic, we have $H(Y, F) = H(Y)$. This leads to:

---

**Definition 3** (Direct sum entropy and KL divergence). *The direct sum entropy of a mixed r.v. $Y$ is*

$$H^\oplus(Y) := H(F) + H(Y \mid F) \tag{10}$$

$$= \underbrace{-\sum_{f \in \bar{\mathcal{F}}(\mathcal{P})} P_F(f) \log P_F(f)}_{\text{discrete entropy}} + \sum_{f \in \bar{\mathcal{F}}(\mathcal{P})} P_F(f) \underbrace{\left( -\int_f p_{Y|F}(\boldsymbol{y} \mid f) \log p_{Y|F}(\boldsymbol{y} \mid f) \right)}_{\text{differential entropy}}.$$

*The KL divergence between distributions $p_Y^\oplus \equiv (P_F, p_{Y|F})$ and $q_Y^\oplus \equiv (Q_F, q_{Y|F})$ is:*

$$D_{\mathrm{KL}}^\oplus(p_Y^\oplus \| q_Y^\oplus) := D_{\mathrm{KL}}(P_F \| Q_F) + \mathbb{E}_{f \sim P_F}\left[ D_{\mathrm{KL}}(p_{Y|F}(\cdot \mid F = f) \| q_{Y|F}(\cdot \mid F = f)) \right] \tag{11}$$

$$= \underbrace{\sum_{f \in \bar{\mathcal{F}}(\mathcal{P})} P_F(f) \log \frac{P_F(f)}{Q_F(f)}}_{\text{discrete KL}} + \sum_{f \in \bar{\mathcal{F}}(\mathcal{P})} P_F(f) \underbrace{\left( \int_f p_{Y|F}(\boldsymbol{y} \mid f) \log \frac{p_{Y|F}(\boldsymbol{y} \mid f)}{q_{Y|F}(\boldsymbol{y} \mid f)} \right)}_{\text{continuous KL}}.$$

---

As shown in Definition 3, the direct sum entropy and the KL divergence have two components: a **discrete one over faces** and an **expectation of a continuous one over each face.** The KL divergence is always non-negative and it becomes $+\infty$ if $\mathrm{supp}(P_F) \not\subseteq \mathrm{supp}(Q_F)$ or if there is some face where $\mathrm{supp}(p_{Y|F=f}) \not\subseteq \mathrm{supp}(q_{Y|F=f})$.[1] App. D provides more information theoretic extensions.

**Relation to optimal codes.** The direct sum entropy and KL divergence have an interpretation in terms of optimal coding, described in Proposition 1 for the case $\mathcal{P} = \triangle_{K-1}$ and proven in App. D. In words, the direct sum entropy is the average length of the optimal code where the sparsity pattern of $\boldsymbol{y} \in \triangle_{K-1}$ must be encoded losslessly and where there is a predefined bit precision for the fractional

---

[1]In particular, this means that mixed distributions shall not be used as a relaxation in VAEs with purely discrete priors using the ELBO – rather, the prior should be also mixed.

entries of $\boldsymbol{y}$. On the other hand, the KL divergence between $p_Y^\oplus$ and $q_Y^\oplus$ expresses the additional average code length if we encode variable $Y \sim p_Y^\oplus(\boldsymbol{y})$ with a code that is optimal for distribution $q_Y^\oplus(\boldsymbol{y})$, and it is independent of the required bit precision.

**Proposition 1.** *Let $Y$ be a mixed r.v. in $\triangle_{K-1}$. In order to encode the face of $Y$ losslessly and to ensure an $N$-bit precise encoding of $Y$ in that face we need the following bits on average:*

$$H_N^\oplus(Y) = H^\oplus(Y) + N \sum_{k=1}^{K}(k-1) \sum_{\substack{f \in \bar{\mathcal{F}}(\triangle_{K-1}) \\ \dim(f)=k-1}} P_F(f). \tag{12}$$

**Entropy of Gaussian-Sparsemax.** Revisiting the Gaussian-Sparsemax with $K = 2$ (§3.3) and using the intrinsic representation (7), the direct sum entropy becomes

$$H(Y) = H(F) + H(Y|F) = -P_0 \log P_0 - P_1 \log P_1 - \int_0^1 \mathcal{N}(y; z, \sigma^2) \log \mathcal{N}(y; z, \sigma^2) \mathrm{d}y. \tag{13}$$

This leads to simple expressions for the entropy and KL divergence, detailed in App. E. We use these expressions in our experiments with mixed bit-vector VAEs in §5.

**Entropy of Mixed Dirichlet.** The intrinsic representation of the Mixed Dirichlet (§3.3) allows for $\mathcal{O}(K)$ computation of $H(F)$ and $D_{\mathrm{KL}}(P_F||Q_F)$ via dynamic programming, necessary for $H^\oplus(Y)$ and $D_{\mathrm{KL}}^\oplus(p_Y^\oplus||q_Y^\oplus)$ (see App. C for details). The continuous parts $H(Y|F)$ and $\mathbb{E}_{P_F}[D_{\mathrm{KL}}(p_{Y|F=f}||q_{Y|F=f})]$ require computing an expectation with an exponential number of terms (one per proper face) and can be approximated with an MC estimate by sampling from $P_F$ and assessing closed-form the differential entropy of Dirichlet distributions over the sampled faces.

**Maximum entropy density in the full simplex.** An important question is to characterize maximum entropy mixed distributions. If we consider only continuous distributions, confined to the maximal face $\mathrm{ri}(\triangle_{K-1})$, the answer is the flat distribution, with entropy $-\log\big((K-1)!\big)$, corresponding to a deterministic $F$ which puts all probability mass in this maximal face. But constraining ourselves to a single face is quite a limitation, and in particular *knowing* this constraint provides valuable information that intuitively should reduce entropy.[2] What if we consider densities that assign probability to the boundaries? This is answered by the next proposition, proved in App. F.

**Proposition 2** (Maxent mixed distribution on simplex). *Let $Y$ be a mixed r.v. on $\triangle_{K-1}$ with $P_F(f) \propto \frac{2^{N(\dim(f))}}{\dim(f)!}$ and $p_{Y|F}(\boldsymbol{y} \mid f)$ uniform for each $f \in \bar{\mathcal{F}}(\triangle_{K-1})$. Then, $Y$ has maximal direct sum entropy $H_{N,\max}^\oplus(Y)$. The value of the entropy is:*

$$H_{N,\max}^\oplus(Y) = \log \sum_{k=1}^{K} \frac{\binom{K}{k} 2^{N(k-1)}}{(k-1)!} = \log L_{K-1}^{(1)}(-2^N). \tag{14}$$

*where $L_n^{(\alpha)}(x)$ denotes the generalized Laguerre polynomial (Sonine, 1880).*

For example, for $K = 2$, the maximal entropy distribution is $P_F(\{0\}) = P_F(\{1\}) = 1/(2+2^N)$, $P_F([0,1]) = 2^N/(2+2^N)$, with $H_{N,\max}^\oplus(Y) = \log(2+2^N)$. Therefore, in the worst case, we need $\log_2(2 + 2^N)$ bits on average to encode $Y \in \triangle_1$ with bit precision $N$. For $K = 3$, $H_{N,\max}^\oplus(Y) = \log(3 + 3 \cdot 2^N + 2^{2N-1})$. A plot of the entropy as a function of $K$ is shown in Figure 4, App. F.

## 5 EXPERIMENTS

We report experiments in three representation learning tasks:[3] an **emergent communication** game, where we assess the ability of mixed latent variables over $\triangle_{K-1}$ to induce sparse communication

---

[2]At the opposite extreme, if we only assign probability to pure vertices, *i.e.*, if we are constrained to *minimal* faces, the maximal discrete entropy is $\log K$. We will see that looking at *all* faces further increases entropy.

[3]Appendix G.4 contains a fourth experiment—regression towards voting proportions—where we use our Mixed Dirichlet as a likelihood function in a generalized linear model.

Table 2: Test results. **Left**: Emergent communication success average and standard error over 10 runs. Random guess baseline: $6.25\%$. **Right**: Fashion-MNIST bit-vector VAE NLL (bits/dim, lower is better). Entropy column legend: continuous/discrete/mixed, $\approx$: estimated, $=$: exact.

| Method | Success (%) | Nonzeros ↓ |
|---|---|---|
| Gumbel-Softmax | 78.84 ±8.07 | 256 |
| Gumbel-Softmax ST | 49.96 ±9.51 | 1 |
| $K$-D Hard Concrete | 76.07 ±7.76 | 21.43 ±17.56 |
| Gaussian-Sparsemax | **80.88** ±0.50 | 1.57 ±0.02 |

| Method | Entropy | NLL | Sparsity (%) ↑ |
|---|---|---|---|
| Binary Concrete | C $\approx$ | 3.60 | 0 |
| Gumbel-Softmax | D $=$ | **3.49** | 0 |
| Gumbel-Softmax ST | D $=$ | 3.57 | 100 |
| Hard Concrete | X $\approx$ | 3.57 | 45.64 |
| Gaussian-Sparsemax | X $\approx$ | 3.53 | 82.82 |
| Gaussian-Sparsemax | X $=$ | **3.49** | 73.83 |

between two agents; a **bit-vector VAE** modeling Fashion-MNIST images (Xiao et al., 2017), where we compare several mixed distributions over $[0, 1]^K$ and study the impact of the direct sum entropy (§4); and a **mixed-latent VAE**, where we experiment with the Mixed Dirichlet over $\triangle_{K-1}$ to model MNIST images (LeCun et al., 2010). Full details about each task and the experimental setup are reported in App. G. Throughout, we report our three mixed distributions described in §3.3 alongside the following distributions: Concrete (Maddison et al. (2017); a purely continuous density), Gumbel-Softmax (Jang et al. (2017); like above, but using a discrete KL term in a VAE, which introduces inconsistency), Gumbel-Softmax ST (Jang et al. (2017); categorical latent variable, but using Concrete straight-through gradients), and Dirichlet and Gaussian, when applicable.

**Emergent communication.** This is a cooperative game between two agents: a *sender* sees an image and emits a single-symbol message from a fixed vocabulary $[K]$; a *receiver* reads the symbol and tries to identify the correct image out of a set. We follow the architecture in Lazaridou & Baroni (2020); Havrylov & Titov (2017), choosing a vocabulary size of 256 and 16 candidate images. For the mixed variable models, the message is a "mixed symbol", *i.e.*, a (sparse) point in $\triangle_{K-1}$. Table 2 reports communication success (accuracy of the receiver), along with the average number of nonzero entries in the samples at test time. Gaussian-Sparsemax learns to become very sparse, attaining the best overall communication success, exhibiting in general a better trade-off than the $K$-D Hard Concrete.

**Bit-Vector VAE.** We model Fashion-MNIST following Correia et al. (2020), with 128 binary latent bits, maximizing the ELBO. Each latent bit has a uniform prior for purely discrete and continuous models, and the maxent prior (Prop. 2), which assigns probability $^1/_3$ to each face of $\triangle_1$, in the mixed case. For Hard Concrete (Louizos et al., 2018) and Gaussian-Sparsemax, we use mixed latent variables and our direct sum entropy, yielding a coherent objective and unbiased gradients. For the Hard Concrete, we estimate $H(Y \mid X = x)$ via MC; for Gaussian-Sparsemax, we consider both cases: using an MC estimate and computing the entropy exactly using the expression derived in §3.3. In Table 2 (right) we report an importance sampling estimate (1024 samples) of negative log-likelihood (NLL) on test data, normalized by the number of pixels. For Gaussian-Sparsemax, exact entropy computation improves performance, matching Gumbel-Softmax while being sparse.

**Mixed-Latent VAE on MNIST.** We model the binarized MNIST dataset using a mixed-latent VAE, with $\boldsymbol{y} \in \Delta_{10-1}$, a maximum-entropy prior on $Y$, and a factorized decoder. We use a feed-forward encoder with a single-hidden-layer to predict the variational distributions (*e.g.*, log-potentials $\boldsymbol{w}$ and concentrations $\boldsymbol{\alpha}$, for Mixed Dirichlet). For Mixed Dirichlet, gradients are estimated via a combination of implicit reparametrization (Figurnov et al., 2018) and the score function estimator (Mnih & Gregor, 2014), see App. C for details. We report a single-sample MC

Table 3: MNIST test results (avg. of 5 runs). Categorical is marginalized exactly.

| Method | D | R | NLL↓ |
|---|---|---|---|
| Gaussian ($\mathbb{R}^{10}$) | 76.67 | 19.94 | 91.12 |
| Dirichlet | 78.62 | 19.94 | 93.81 |
| Categorical | 164.72 | 2.28 | 166.95 |
| Gumbel-Softmax ST | 171.76 | 1.70 | 168.50 |
| Mixed Dirichlet | 90.34 | 19.39 | 106.59 |

estimate of distortion (D) and rate (R) as well as a 1000-samples importance sampling estimate of NLL. The Gaussian prior has access to all of the $\mathbb{R}^{10}$, whereas all other priors are constrained to the simplex $\Delta_{10-1}$, namely, its relative interior (Dirichlet), its vertices (Gumbel-Softmax ST, and Categorical), or all of it (Mixed Dirichlet). The Dirichlet model clusters digits as well as a Gaussian VAE does, while purely discrete models struggle to discover structure. App. G lists qualitative evidence. Compared to a Dirichlet, the Mixed Dirichlet makes some plausible confusions (Fig. 3,

left), but, crucially, it solves part of the problem by allocating digits to specific faces (Fig. 3, right). Even though we employed a sampled self-critic, Mixed Dirichlet seems to suffer from variance of SFE, which may explain why Mixed Dirichlet under-performs in terms of NLL.

# 6 RELATED WORK

Existing strategies to learn discrete latent variable models include the score function estimator (Williams, 1992; Mnih & Gregor, 2014) and pathwise gradients combined with a Concrete relaxation (Maddison et al., 2017; Jang et al., 2017). The latter is often used with straight-through gradients and by combining a continuous latent variable with a discrete entropy, which is theoretically not sound. Besides the Concrete, continuous versions of the Bernoulli and Categorical have also been proposed (Loaiza-Ganem & Cunningham, 2019; Gordon-Rodriguez et al., 2020), but they are also purely continuous densities, assigning zero probability mass to the boundary of the simplex.

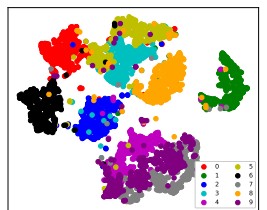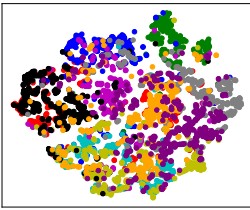

Figure 3: tSNE plots of: posterior samples $y \in \triangle_{K-1}$ (left), predicted log-potentials $\boldsymbol{w} \in \mathbb{R}^K$ (right). Colors encode digit label (not available to models). Clusters are formed in latent space and in how digits are assigned to faces (recall that $\boldsymbol{w}$ parameterizes a Gibbs distribution over $\bar{\mathcal{F}}(\triangle_{K-1})$).

Our approach is inspired by discrete-continuous hybrids based on truncation and rectification (Hinton & Ghahramani, 1997; Palmer et al., 2017; Louizos et al., 2018), which have been proposed for univariate distributions. We generalize this idea to arbitrary dimensions replacing truncation by sparse projections to the simplex. The direct sum measure and our proposed intrinsic sampling strategies are related to the concept of "manifold stratification" proposed in the statistical physics literature (Holmes-Cerfon, 2020). Other mixed r.vs. have also been recently considered (for a few special cases) in (Bastings et al., 2019; Murady, 2020; van der Wel, 2020). For instance, Burkhardt & Kramer (2019) induce sparsity in hierarchical models by deterministically masking the concentration parameters of a Dirichlet distribution yielding a special case of Mixed Dirichlet with degenerate $P_F$, trained with the straight-through estimator. Our paper generalizes these attempts, providing solid theoretical support for manipulating mixed distributions in higher dimensions.

Discrete and continuous representations in the context of emergent communication have been discussed in Foerster et al. (2016); Lazaridou & Baroni (2020). Discrete communication is computationally more challenging because it prevents direct gradient backpropagation (Foerster et al., 2016; Havrylov & Titov, 2017), but it is hypothesized that this "discrete bottleneck" forces the emergence of symbolic protocols. Our mixed variables bring a new perspective into this problem, leading to *sparse communication*, which lies in between discrete and continuous communication and supports gradient backpropagation without the need for straight-through gradients.

# 7 CONCLUSIONS

We presented a mathematical framework for handling mixed random variables, which are discrete/continuous hybrids. Key to our framework is the use of a direct sum measure as an alternative to the Lebesgue-Borel and the counting measures, which considers all faces of the simplex. We developed generalizations of information theoretic concepts for mixed symbols, and we experimented on emergent communication and on variational modeling of MNIST and Fashion-MNIST images.

We believe the framework described here is only scratching the surface. For example, the Mixed Dirichlet is just one example of an intrinsic mixed distribution; more effective intrinsic parametrizations may exist and are a promising avenue. While our main focus was on the probability simplex and hypercube, mixed *structured* variables are another promising direction, enabled by our theoretical characterization of direct sum measures, which can be defined for any polytope via their face lattice (Ziegler, 1995; Grünbaum, 2003). Current methods for structured variables include perturbations (continuous, Corro & Titov (2019); Berthet et al. (2020); Paulus et al. (2020)), and sparsemax extensions (discrete, Niculae et al. (2018); Correia et al. (2020)), lacking tractable densities.

**Ethics statement.** We highlight indirect impact of our work through applications such as generation and explainable AI, where improved performance must be carefully scrutinized. Our proof-of-concept emergent communication, like previous work, uses ImageNet, whose construction exhibits societal biases, including racism and sexism (Crawford & Paglen, 2019) that communicating agents may learn. Even in unsupervised settings, biases may be learned by communicating agents.

**Reproducibility statement.** We now discuss the efforts that have been made to ensure reproducibility of our work. We state the full set of assumptions of our theoretical results and include complete proofs in App. A, App. B, App. C, App. D, App. E, and App. F. Additionally, code and instructions to reproduce our experiments are available at `https://github.com/deep-spin/sparse-communication`. We report the standard error over 10 runs for the emergent communication experiment due to the high variance of results across seeds. We include the type of computing resources used in our experiments in App. G.5.

## ACKNOWLEDGMENTS

We would like to thank Mário Figueiredo, Gonçalo Correia, and the DeepSPIN team for helpful discussions, Tim Vieira, who answered several questions about order statistics, Sam Power, who pointed out to manifold stratification, and Juan Bello-Rivas, who suggested the name "mixed random variables." This work was built on open-source software; we acknowledge Van Rossum & Drake (2009); Oliphant (2006); Virtanen et al. (2020); Walt et al. (2011); Pedregosa et al. (2011), and Paszke et al. (2019). AF and AM are supported by the P2020 program MAIA (LISBOA-01-0247- FEDER-045909), the European Research Council (ERC StG DeepSPIN 758969), and by the Fundação para a Ciência e Tecnologia through project PTDC/CCI-INF/4703/2021 (PRELUNA) and contract UIDB/50008/2020. WA received funding from the European Union's Horizon 2020 research and innovation programme under grant agreement No 825299 (GoURMET). VN is partially supported by the Hybrid Intelligence Centre, a 10-year programme funded by the Dutch Ministry of Education, Culture and Science through the Netherlands Organisation for Scientific Research (`https://hybrid-intelligence-centre.nl`).

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

## A   PROOF OF WELL-DEFINEDNESS OF DIRECT SUM MEASURE

We start by recalling the definitions of $\sigma$-algebras, measures, and measure spaces. A $\sigma$**-algebra** on a set $X$ is a collection of subsets, $\Omega \subseteq 2^X$, which is closed under complements and under countable unions. A **measure** $\mu$ on $(X, \Omega)$ is a function from $\Omega$ to $\mathbb{R} \cup \{\pm\infty\}$ satisfying (i) $\mu(A) \geq 0$ for all $A \in \Omega$, (ii) $\mu(\varnothing) = 0$, and (iii) the $\sigma$-additivity property: $\mu(\sqcup_{j \in \mathbb{N}} A_j) = \sum_{j \in \mathbb{N}} \mu(A_j)$ for every countable collections $\{A_j\}_{j \in \mathbb{N}} \subseteq \Omega$ of pairwise disjoint sets in $\Omega$. A **measure space** is a triple $(X, \Omega, \mu)$ where $X$ is a set, $\mathcal{A}$ is a $\sigma$-algebra on $X$ and $\mu$ is a measure on $(X, \mathcal{A})$. An example is the Euclidean space $X = \mathbb{R}^K$ endowed with the Lebesgue measure, where $\Omega$ is the Borel algebra generated by the open sets (*i.e.* the set $\Omega$ which contains these open sets and countably many Boolean operations over them).

The well-definedness of the direct sum measure $\mu^\oplus$ comes from the following more general result, which appears (without proof) as exercise I.6 in Conway (2019).

**Lemma 1.** *Let $(X_k, \Omega_k, \mu_k)$ be measure spaces for $k = 1, \ldots, K$. Then, $(X, \Omega, \mu)$ is also a measure space, with $X = \bigoplus_{k=1}^K X_k = \prod_{k=1}^K X_k$ (the direct sum or Cartesian product of sets $X_k$), $\Omega = \{A \subseteq X \mid A \cap X_k \in \Omega_k, \ \forall k \in [K]\}$, and $\mu(A) = \sum_{k=1}^K \mu_k(A \cap X_k)$.*

*Proof.* First, we show that $\Omega$ is a $\sigma$-algebra. We need to show that (i) if $A \in \Omega$, then $\bar{A} \in \Omega$, and (ii) if $A_i \in \Omega$ for each $i \in \mathbb{N}$ then $\bigcup_{i \in \mathbb{N}} A_i \in \Omega$. For (i), we have that, if $A \in \Omega$, then we must have $A \cap X_k \in \Omega_k$ for every $k$, and therefore $\bar{A} \cap X_k = X_k \setminus A = X_k \setminus (A \cap X_k) \in \Omega_k$, since $\Omega_k$ is a $\sigma$-algebra on $X_k$. This implies that $\bar{A} \in \Omega$. For (ii), we have that, if $A_i \in \Omega$, then we must have $A_i \cap X_k \in \Omega_k$ for every $i \in \mathbb{N}$ and $k \in [K]$, and therefore $\left(\bigcup_{i \in \mathbb{N}} A_i\right) \cap X_k = \bigcup_{i \in \mathbb{N}} (A_i \cap X_k) \in \Omega_k$, since $\Omega_k$ is closed under countable unions. This implies that $\bigcup_{i \in \mathbb{N}} A_i \in \Omega$. Second, we show that $\mu$ is a measure. We clearly have $\mu(A) = \sum_{k=1}^K \mu_k(A \cap X_k) \geq 0$, since each $\mu_k$ is a measure itself, and hence it is non-negative. We also have $\mu(\varnothing) = \sum_{k=1}^K \mu_k(\varnothing \cap X_k) = \sum_{k=1}^K \mu_k(\varnothing) = 0$. Finally, if $\{A_j\}_{j \in \mathbb{N}} \subseteq \Omega$ is a countable collection of disjoint sets, we have $\mu(\sqcup_{j \in \mathbb{N}} A_j) = \sum_{k=1}^K \mu_k(\sqcup_{j \in \mathbb{N}} (A_j \cap X_k)) = \sum_{k=1}^K \sum_{j \in \mathbb{N}} \mu_k(A_j \cap X_k) = \sum_{j \in \mathbb{N}} \sum_{k=1}^K \mu_k(A_j \cap X_k) = \sum_{j \in \mathbb{N}} \mu(A_j)$. $\qquad\square$

We have seen in §3 that the simplex $\triangle_{K-1}$ can be decomposed as a disjoint union of the relative interior of its faces. Each of these relative interiors is an open subset of an affine subspace isomorphic to $\mathbb{R}^{k-1}$, for $k \in [K]$, which is equipped with the Lebesgue measure for $k > 1$ and the counting measure for $k = 1$. Lemma 1 then guarantees that we can take the direct sum of all these affine spaces as a measure space with the direct sum measure $\mu = \mu^\oplus$ of Definition 1.

## B   $K$-DIMENSIONAL GAUSSIAN-SPARSEMAX

We derive expressions for the density $p_{Y,F}(y, f)$ for the sample-and-project case (stochastic sparsemax). We assume without loss of generality that $f = \{1, 2, \ldots, s\}$ and that we want to compute the density $p_{Y_{2:K}}(y_2, \ldots, y_s, 0, \ldots, 0)$ for given $y_2, \ldots, y_s > 0$ such that $y_1 := 1 - \sum_{j=2}^s y_j > 0$.

The process that generates the data is as follows: first, $z_i \sim p_{Z_i}(z_i)$ independently for $i \in [K]$. Then, $y$ is obtained deterministically from $z_{1:K}$ as $y = \text{sparsemax}(z_{1:K})$. We assume here that each $p_{Z_i}(z_i)$ is a univariate Gaussian distribution with mean $\mu_i$ and variance $\sigma_i^2$.

We make use of the following well-known properties of multivariate Gaussians:

**Lemma 2.** *Let $Z \sim \mathcal{N}(\mu, \Sigma)$. Then, for any matrix $A$, not necessarily square we have $AZ \sim \mathcal{N}(A\mu, A\Sigma A^\top)$. Furthermore, splitting*

$$Z = \begin{bmatrix} X \\ Y \end{bmatrix}, \quad \mu = \begin{bmatrix} \mu_x \\ \mu_y \end{bmatrix}, \quad \Sigma = \begin{bmatrix} \Sigma_{xx} & \Sigma_{xy} \\ \Sigma_{xy}^\top & \Sigma_{yy} \end{bmatrix}, \tag{15}$$

*we have the following expression for the marginal distribution $P_X(x)$:*

$$X \sim \mathcal{N}(\mu_x, \Sigma_{xx}) \tag{16}$$

*and the following expression for the conditional distribution $P_{X|Y}(x \mid Y = y)$:*

$$X \mid Y = y \sim \mathcal{N}(\mu_x + \Sigma_{xy}\Sigma_{yy}^{-1}(y - \mu_y), \Sigma_{xx} - \Sigma_{xy}\Sigma_{yy}^{-1}\Sigma_{xy}^\top). \tag{17}$$

We start by picking one index in the support of $y$ – we assume without loss of generality that this pivot index is 1 and that the support set is $\{1, \ldots, s\}$ as stated above – and introducing new random variables $U_i := Z_i - Z_1$ for $i \in \{2, \ldots, K\}$. Note that these new random variables are not independent since they all depend on $Z_1$. Then, from the change of variable formula for non-invertible transformations, we have, for given $y_2, \ldots, y_s > 0$ such that $y_1 := 1 - \sum_{j=2}^s y_j > 0$:

$$p_{Y_{2:K}}(y_2, \ldots, y_s, 0, \ldots, 0) = p_{U_{2:s}}(u_{2:s}) \times |B| \times$$
$$\int_{-\infty}^{-y_1} \cdots \int_{-\infty}^{-y_1} p_{U_{(s+1):K}|U_{1:s}}(u_{(s+1):K} \mid U_{1:s} = u_{1:s}) du_{s+1} \cdots du_K. \tag{18}$$

where $u_i = y_i - y_1$ for $i \in 2, \ldots, s$, which can written as $u_{2:s} = By_{2:s} + c$ with $B = I_{s-1} - 1_{s-1}1_{s-1}^\top$ and $c = -1_{s-1}$. The determinant of $B$ is simply $|B| = s$.

Note that $U_{2:K} = AZ_{1:K}$, where $A = [-1_{K-1}, I_{K-1}] \in \mathbb{R}^{(K-1) \times K}$, that is

$$A = \begin{bmatrix} -1 & 1 & 0 & \cdots & 0 \\ -1 & 0 & 1 & \cdots & 0 \\ \vdots & & & \ddots & 0 \\ -1 & 0 & 0 & \cdots & 1 \end{bmatrix}. \tag{19}$$

From Lemma 2, we have that

$$p_U(u) = \mathcal{N}(u; A\mu, A\mathrm{Diag}(\sigma^2)A^\top)$$
$$= \mathcal{N}(u; \mu_{2:K} - \mu_1 1_{K-1}, \mathrm{Diag}(\sigma_{2:K}^2) + \sigma_1^2 1_{K-1}1_{K-1}^\top), \tag{20}$$

and the marginal distribution $p_{U_{2:s}}(u_{2:s})$ is

$$p_{U_{2:s}}(u_{2:s}) = \mathcal{N}(u_{2:s}; \mu_{2:s} - \mu_1 1_{s-1}, \mathrm{Diag}(\sigma_{2:s}^2) + \sigma_1^2 1_{s-1}1_{s-1}^\top)$$
$$= \mathcal{N}(y_{2:s} - y_1 1_{s-1}; \mu_{2:s} - \mu_1 1_{s-1}, \mathrm{Diag}(\sigma_{2:s}^2) + \sigma_1^2 1_{s-1}1_{s-1}^\top). \tag{21}$$

Note that this is a multivariate distribution whose covariance matrix is the sum of a diagonal matrix with a constant matrix.

We now calculate the conditional distribution of the variables which are *not* in the support conditioned on the ones which *are* in the support. We have according to the notation in Lemma 2:

$$\Sigma_{xx} = \mathrm{Diag}(\sigma_{(s+1):K}^2) + \sigma_1^2 1_{K-s}1_{K-s}^\top$$
$$\Sigma_{yy} = \mathrm{Diag}(\sigma_{2:s}^2) + \sigma_1^2 1_{s-1}1_{s-1}^\top$$
$$\Sigma_{xy} = \sigma_1^2 1_{K-s}1_{s-1}^\top. \tag{22}$$

Using the Sherman-Morrison formula, we obtain

$$\Sigma_{yy}^{-1} = \mathrm{Diag}(\sigma_{2:s}^{-2}) - \frac{\sigma_{2:s}^{-2}(\sigma_{2:s}^{-2})^\top}{\sigma_1^{-2} + \sum_{i=2}^s \sigma_i^{-2}}, \tag{23}$$

from which we get

$$\tilde{\mu} := \mu_x + \Sigma_{xy}\Sigma_{yy}^{-1}(y - \mu_y)$$
$$= \mu_{(s+1):K} - \mu_1 1_{K-s} + \frac{1}{\sigma_1^{-2} + \sum_{i=2}^s \sigma_i^{-2}} 1_{K-s}(\sigma_{2:s}^{-2})^\top(u_{2:s} - \mu_{2:s} + \mu_1 1_{s-1})$$
$$= \mu_{(s+1):K} - \mu_1 1_{K-s} + \frac{1}{\sum_{i=1}^s \sigma_i^{-2}} 1_{K-s}(\sigma_{2:s}^{-2})^\top(y_{2:s} - y_1 1_{s-1} - \mu_{2:s} + \mu_1 1_{s-1})$$
$$= \mu_{(s+1):K} - y_1 1_{K-s} + \frac{\sum_{i=1}^s \sigma_i^{-2}(y_i - \mu_i)}{\sum_{i=1}^s \sigma_i^{-2}} 1_{K-s}. \tag{24}$$

and

$$\tilde{\Sigma} := \Sigma_{xx} - \Sigma_{xy}\Sigma_{yy}^{-1}\Sigma_{xy}^\top$$

$$= \mathrm{Diag}(\sigma_{(s+1):K}^2) + \sigma_1^2 1_{K-s} 1_{K-s}^\top - \frac{\sigma_1^2 1_{K-s}(\sigma_{2:s}^{-2})^\top 1_{s-1} 1_{K-s}^\top}{\sigma_1^{-2} + \sum_{i=2}^s \sigma_i^{-2}}$$

$$= \mathrm{Diag}(\sigma_{(s+1):K}^2) + \sigma_1^2 1_{K-s} 1_{K-s}^\top - \frac{\sigma_1^2 \sum_{i=2}^s \sigma_i^{-2}}{\sigma_1^{-2} + \sum_{i=2}^s \sigma_i^{-2}} 1_{K-s} 1_{K-s}^\top$$

$$= \mathrm{Diag}(\sigma_{(s+1):K}^2) + \sigma_1^2 \left( 1 - \frac{\sum_{i=2}^s \sigma_i^{-2}}{\sigma_1^{-2} + \sum_{i=2}^s \sigma_i^{-2}} \right) 1_{K-s} 1_{K-s}^\top$$

$$= \mathrm{Diag}(\sigma_{(s+1):K}^2) + \frac{1}{\sigma_1^{-2} + \sum_{i=2}^s \sigma_i^{-2}} 1_{K-s} 1_{K-s}^\top$$

$$= \mathrm{Diag}(\sigma_{(s+1):K}^2) + \frac{1}{\sum_{i=1}^s \sigma_i^{-2}} 1_{K-s} 1_{K-s}^\top. \tag{25}$$

Finally, also from Lemma 2, we get

$$p_{U_{(s+1):K}|U_{1:s}}(u_{(s+1):K} \mid U_{1:s} = u_{1:s}) = \mathcal{N}(u_{(s+1):K}; \tilde{\mu}, \tilde{\Sigma}). \tag{26}$$

Note that this is again a multivariate Gaussian distribution with a covariance matrix $\tilde{\Sigma}$ which is the sum of a diagonal and a constant matrix.

Putting everything together, we get

$$p_{Y_{2:K}}(y_2, \ldots, y_s, 0, \ldots, 0) = s\mathcal{N}(y_{2:s} - y_1 1_{s-1}; \mu_{2:s} - \mu_1 1_{s-1}, \mathrm{Diag}(\sigma_{2:s}^2) + \sigma_1^2 1_{s-1} 1_{s-1}^\top) \times$$

$$F \left( \mu_{(s+1):K} + \frac{\sum_{i=1}^s \sigma_i^{-2}(y_i - \mu_i)}{\sum_{i=1}^s \sigma_i^{-2}} 1_{K-s}; \tilde{\Sigma} \right), \tag{27}$$

where $F(v; \tilde{\Sigma}) = \int_{-\infty}^0 \cdots \int_{-\infty}^0 \mathcal{N}(0; v, \tilde{\Sigma})$ is the negative orthant cumulative distribution of a multivariate Gaussian with mean $v$ and covariance $\tilde{\Sigma}$. Efficient Monte Carlo approximations of this integral have been proposed by Genz (1992). Using again Lemma 2 but now in the reverse direction, the function $F(v; \tilde{\Sigma})$, setting $v = \mu_{(s+1):K} + \frac{\sum_{i=1}^s \sigma_i^{-2}(y_i - \mu_i)}{\sum_{i=1}^s \sigma_i^{-2}} 1_{K-s}$, can be reduced to a unidimensional integral:

$$F(v; \tilde{\Sigma}) = \int_{-\infty}^\infty \mathrm{d}t \, \mathcal{N}\left( t; \frac{\sum_{i=1}^s \sigma_i^{-2}(\mu_i - y_i)}{\sum_{i=1}^s \sigma_i^{-2}}, \frac{1}{\sum_{i=1}^s \sigma_i^{-2}} \right) \prod_{i=s+1}^K \int_{-\infty}^t \mathrm{d}r \mathcal{N}(r; \mu_i, \sigma_i^2)$$

$$= \int_{-\infty}^\infty \mathrm{d}t \, \phi\left( \frac{\sum_{i=1}^s \sigma_i^{-2}(t + y_i - \mu_i)}{\sqrt{\sum_{i=1}^s \sigma_i^{-2}}} \right) \prod_{i=s+1}^K \Phi\left( \frac{t - \mu_i}{\sigma_i} \right)$$

$$= \int_0^1 \mathrm{d}u \prod_{j=s+1}^K \Phi\left( \frac{\Phi^{-1}(u)}{\sigma_j \sqrt{\sum_{i=1}^s \sigma_i^{-2}}} - \frac{\sum_{i=1}^s \sigma_i^{-2}(y_i - \mu_i + \mu_j)}{\sigma_j \sum_{i=1}^s \sigma_i^{-2}} \right). \tag{28}$$

where $\phi(x)$ and $\Phi(x)$ are the p.d.f. and c.d.f. of a standard Gaussian distribution, respectively, and we applied the change of variables formula in the last line, with $u = \Phi\left( \sum_{i=1}^s \sigma_i^{-2}(t + y_i - \mu_i)/\sqrt{\sum_{i=1}^s \sigma_i^{-2}} \right)$, whose inverse is $t = \frac{\Phi^{-1}(u)}{\sqrt{\sum_{i=1}^s \sigma_i^{-2}}} - \frac{\sum_{i=1}^s \sigma_i^{-2}(y_i - \mu_i)}{\sum_{i=1}^s \sigma_i^{-2}}$; the Jacobian of this transformation cancels with the $\phi$-term. We can compute the quantities above via the erf function and its inverse $\mathrm{erf}^{-1}$, as $\Phi(x) = \frac{1}{2}\left( 1 + \mathrm{erf}\left( \frac{x}{\sqrt{2}} \right) \right)$ and $\Phi^{-1}(u) = \sqrt{2}\mathrm{erf}^{-1}(2u - 1)$.

When the variance is constant, $\sigma_i = \sigma \ \forall i \in [K]$, and since $\sum_{i=1}^{s} y_i = 1$, the expressions above simplify to

$$
\begin{aligned}
F(v; \tilde{\Sigma}) &= \int_{-\infty}^{\infty} \mathrm{d}t \ \mathcal{N}\left(t; \frac{-1 + \sum_{i=1}^{s} \mu_i}{|s|}, \frac{\sigma^2}{|s|}\right) \prod_{i=s+1}^{K} \int_{-\infty}^{t} \mathrm{d}r \mathcal{N}(r; \mu_i, \sigma^2) \\
&= \int_{-\infty}^{\infty} \mathrm{d}t \ \phi\left(\frac{1 + |s|t - \sum_{i=1}^{s} \mu_i}{\sigma\sqrt{|s|}}\right) \prod_{i=s+1}^{K} \Phi\left(\frac{t - \mu_i}{\sigma}\right) \\
&= \int_{0}^{1} \mathrm{d}u \prod_{j=s+1}^{K} \Phi\left(\frac{\Phi^{-1}(u)}{\sqrt{|s|}} - \frac{1}{\sigma}\left(\mu_j + \frac{1 - \sum_{i=1}^{s} \mu_i}{|s|}\right)\right).
\end{aligned}
\tag{29}
$$

## C  SAMPLING FROM THE MIXED DIRICHLET DISTRIBUTION

PyTorch code for a batched implementation of the procedures discussed in this section will be made available online.

**Distribution over Face Lattice.**  The distribution $P_F$ is a discrete exponential family whose support is the set of all $2^K - 1$ proper faces of the simplex. The computation of the natural parameter in Equation (8) factorizes over vertices, but, because the empty face is not in the support of $P_F$, the computation of the log-normalizer requires special attention. The obvious strategy is to enumerate all assignments to $(F_{\mathcal{I}}, \boldsymbol{w}^\top \boldsymbol{\phi}(F_{\mathcal{I}}))$, one per proper face. This strategy is not feasible for even moderately large $K$. We can, however, exploit a compact representation of the set of proper faces by encoding the face lattice in a directed acyclic graph (DAG) where each proper face is associated with a complete path from the DAG's source to the DAG's sink.

This DAG $\mathcal{G} = (\mathcal{Q}, \mathcal{A})$ is a collection of states $u \in \mathcal{Q} \subset \{0, \dots, K+1\} \times \{0, 1\} \times \{0, 1\}$ and arcs $(u, v) \in \mathcal{A} \subseteq \mathcal{Q} \times \mathcal{Q}$. A path that ends in $(k, b, s) \in \mathcal{Q}$ corresponds to a non-empty subset of vertices if, and only if, $s = 1$. A path that contains $(k, b, s) \in \mathcal{Q}$ corresponds to a face that includes the $k$th vertex if, and only if, $b = 1$. If $u = (k_1, b_1, s_1) \in \mathcal{Q}$ and $v = (k_2, b_2, s_2) \in \mathcal{Q}$, then $u \leq v$ iff $k_1 \leq k_2$. The state $(0, 0, 0)$ is the DAG's unique source, and the state $(K+1, 0, 1)$ is the DAG's unique sink. Because $= 1$ for the sink, no complete path (*i.e.*, from source to sink) will correspond to an empty face. An arc is a pair of states $(u, v)$, where $u$ is the origin and $v$ is the destination. For every state $(k, b, s)$ such that $k < K$, we have arcs $((k, b, s), (k+1, 1, 1))$ and $((k, b, s), (k+1, 0, s))$. For every state $(K, b, 1)$, we have an arc to the final state $(K+1, 0, 1)$. By construction, the number of states (and arcs) in the DAG is proportional to $K$. A complete path (from source to sink) has length $K + 1$, and it uniquely identifies a proper face. To compute the log-normalizer of Equation (8), we run the forward algorithm through $\mathcal{G}$, with an arc's weight given by $(-1^{1-b})w_k$ if the arc's destination is the state $(k, b, s)$ or 0 if the arc's destination is the DAG's sink. Running the backward algorithm through $\mathcal{G}$ (that is, running the forward algorithm from sink to source) evaluates the marginal probabilities required for ancestral sampling.

**Entropy of Mixed Dirichlet.**  The intrinsic representation of a Mixed Dirichlet distribution allows for efficient computation of $H(F)$ and $D_{\mathrm{KL}}(P_F \| Q_F)$ necessary for $H^{\oplus}(Y)$ and $D_{\mathrm{KL}}^{\oplus}(p_Y^{\oplus} \| q_Y^{\oplus})$. The continuous parts $H(Y|F)$ and $\mathbb{E}_{P_F}[D_{\mathrm{KL}}(p_{Y|F=f} \| q_{Y|F=f})]$ require solving an expectation with an exponential number of terms (one per proper face). The distribution $P_F$ is a discrete exponential family indexed by the natural parameter $\boldsymbol{w} \in \mathbb{R}^K$, its discrete entropy is $H(F) = \log Z(\boldsymbol{w}) - \langle \boldsymbol{w}, \nabla_{\boldsymbol{w}} \log Z(\boldsymbol{w}) \rangle$, where $\nabla_{\boldsymbol{w}} \log Z(\boldsymbol{w}) = \mathbb{E}[\boldsymbol{\phi}(F)]$. KL divergence from a member $Q_F$ of the same family but with parameter $\boldsymbol{v}$ is given by

$$
D_{\mathrm{KL}}(P_F \| Q_F) = \log Z(\boldsymbol{v}) - \log Z(\boldsymbol{w}) - \langle \boldsymbol{v} - \boldsymbol{w}, \nabla_{\boldsymbol{w}} \log Z(\boldsymbol{w}) \rangle.
\tag{30}
$$

The forward-backward algorithm computes both the log-normalizer and its gradient (the expected sufficient statistic) in a single pass through a DAG of size $\mathcal{O}(K)$. For results concerning entropy, cross-entropy, and relative entropy of exponential families see for example Nielsen & Nock (2010).

The continuous part $H(Y|F)$ is the expectation of the differential entropy of $Y|F = f$, each a Dirichlet distribution over the face $f$, under $P_F$. For small $K$ we can enumerate the $2^K - 1$ terms in this expectation, since the entropy of each Dirichlet is known in closed-form. In general, for large $K$,

we can obtain an MC estimate by sampling independently from $P_F$ and assessing the differential entropy of $Y|F = f$ only for sampled faces. For $\mathbb{E}_{P_F}[D_{\mathrm{KL}}(p_{Y|F=f}||q_{Y|F=f})]$ the situation is similar, since KL for two Dirichlet distributions on the same face $f$ is known in closed-form.

**Gradient Estimation.** Parametrizing a Mixed Dirichlet variable takes two parameter vectors, namely, $\boldsymbol{w}$ and $\boldsymbol{\alpha}$. In a VAE, those are predicted from a given data point (*e.g.*, an MNIST digit) by an inference network. To update the parameters of the inference network we need a Monte Carlo estimate of the gradient $\nabla_{\boldsymbol{w},\boldsymbol{\alpha}}\mathbb{E}_{F|\boldsymbol{w}}[\mathbb{E}_{Y|F=f,\boldsymbol{\alpha}}[\ell(y)]]$ with respect to $\boldsymbol{w}$ and $\boldsymbol{\alpha}$ of a loss function $\ell$ computed in expectation under the Mixed Dirichlet distribution. By chain rule, we can rewrite the gradient as follows:

$$\mathbb{E}_{F|\boldsymbol{w}}\left[\mathbb{E}_{Y|F=f,\boldsymbol{\alpha}}[\ell(y)]\nabla_{\boldsymbol{w}}\log P_F(f|\boldsymbol{w}) + \nabla_{\boldsymbol{\alpha}}\mathbb{E}_{Y|F=f,\boldsymbol{\alpha}}[\ell(y)]\right] . \tag{31}$$

Given a sampled face $F = f$, we can MC estimate both contributions to the gradient, the first takes MC estimation of $\mathbb{E}_{Y|F=f',\boldsymbol{\alpha}}[\ell(y)]$, which is straightforward, the second can be done via implicit reprametrization (Figurnov et al., 2018). To reduce the variance of the score function estimator (first term), we employ a sampled self-critic baseline, *i.e.*, an MC estimate of $\mathbb{E}_{Y|F=f',\boldsymbol{\alpha}}[\ell(y)]$ given an independently sampled face $f'$.

## D INFORMATION THEORY FOR MIXED RANDOM VARIABLES

### D.1 MUTUAL INFORMATION FOR MIXED RANDOM VARIABLES

Besides the direct sum entropy and the Kullback-Leibler divergence for mixed distributions, we can also define a mutual information for mixed random variables as follows:

**Definition 4** (Mutual information). *For mixed random variables $Y$ and $Z$, the mutual information between $Y$ and $Z$ is*

$$\begin{aligned} I^{\oplus}(Y;Z) &= H^{\oplus}(Y) - H^{\oplus}(Y \mid Z) \\ &= H(F) + H(Y \mid F) - H(F \mid Z) + H(Y \mid F, Z) \\ &= I(F;Z) + I(Y;Z \mid F) \geq 0. \end{aligned} \tag{32}$$

With these ingredients it is possible to provide counterparts for channel coding theorems by combining Shannon's discrete and continuous channel theorems (Shannon, 1948).

### D.2 PROOF OF PROPOSITION 1 (CODE OPTIMALITY)

Proposition 1 is a consequence of the following facts (Shannon, 1948): The discrete entropy of a random variable representing an alphabet symbol corresponds to the average length of the optimal code for the symbols in the alphabet, in a lossless compression setting. Besides, it is known (Cover & Thomas, 2012) that the optimal number of bits to encode a $D$-dimensional continuous random variable with $N$ bits of precision equals its differential entropy (in bits) plus $ND$.[4] Therefore, the direct sum entropy (10) is the average length of the optimal code where the sparsity pattern of $\boldsymbol{y} \in \triangle_{K-1}$ must be encoded losslessly and where there is a predefined bit precision for the fractional entries of $\boldsymbol{y}$. Eq. 12 follows from these facts and the definition of direct sum entropy (Definition 3).

---

[4]Cover & Thomas (2012, Theorem 9.3.1) provide an informal proof for $D = 1$, but it is straightforward to extend the same argument for $D > 1$.

# E DIRECT SUM ENTROPY AND KULLBACK-LEIBLER DIVERGENCE OF 2D GAUSSIAN-SPARSEMAX

From (1) and Definition 3, the direct sum entropy of the 2D Gaussian sparsemax is:

$$
\begin{aligned}
H^{\oplus}(Y) &= H(F) + H(Y \mid F) \\
&= H([P_0, P_1, 1 - P_0 - P_1]) - (1 - P_0 - P_1) \int_0^1 \frac{\mathcal{N}(y; z, \sigma^2)}{1 - P_0 - P_1} \log \frac{\mathcal{N}(y; z, \sigma^2)}{1 - P_0 - P_1} dy \\
&= -P_0 \log P_0 - P_1 \log P_1 - \int_0^1 \mathcal{N}(y; z, \sigma^2) \log \mathcal{N}(y; z, \sigma^2) dy,
\end{aligned}
\tag{33}
$$

where we have $P_0 = \frac{1 - \operatorname{erf}(z/(\sqrt{2}\sigma))}{2}$, $P_1 = \frac{1 + \operatorname{erf}((z-1)/(\sqrt{2}\sigma))}{2}$, and

$$
\begin{aligned}
&- \int_0^1 \mathcal{N}(y; z, \sigma^2) \log \mathcal{N}(y; z, \sigma^2) dy = \\
&= \int_0^1 \mathcal{N}(y; z, \sigma^2) \left( \log(\sqrt{2\pi\sigma^2}) + \frac{(y-z)^2}{2\sigma^2} \right) dy \\
&= (1 - P_0 - P_1) \log(\sqrt{2\pi\sigma^2}) + \frac{\sigma}{2} \int_{\frac{-z}{\sigma}}^{\frac{1-z}{\sigma}} \mathcal{N}(t; 0, 1) t^2 dt \\
&= (1 - P_0 - P_1) \log(\sqrt{2\pi\sigma^2}) \\
&\quad + \frac{\sigma}{2} \left( \frac{\operatorname{erf}\left(\frac{1-z}{\sqrt{2\sigma^2}}\right) - \operatorname{erf}\left(-\frac{z}{\sqrt{2\sigma^2}}\right)}{2} - \frac{1-z}{\sigma} \mathcal{N}\left(\frac{1-z}{\sigma}; 0, 1\right) - \frac{z}{\sigma} \mathcal{N}\left(-\frac{z}{\sigma}; 0, 1\right) \right),
\end{aligned}
\tag{34}
$$

which leads to a closed form for the entropy. As for the KL divergence, we have:

$$
\begin{aligned}
D_{\mathrm{KL}}^{\oplus}(p_Y^{\oplus} \| q_Y^{\oplus}) &:= D_{\mathrm{KL}}(P_F \| Q_F) + \mathbb{E}_{f \sim P_F} \left[ D_{\mathrm{KL}}(p_{Y|F}(\cdot \mid F = f) \| q_{Y|F}(\cdot \mid F = f)) \right] \\
&= \sum_{f \in \bar{\mathcal{F}}(\mathcal{P})} P_F(f) \log \frac{P_F(f)}{Q_F(f)} + \sum_{f \in \bar{\mathcal{F}}(\mathcal{P})} P_F(f) \left( \int_f p_{Y|F}(\boldsymbol{y} \mid f) \log \frac{p_{Y|F}(\boldsymbol{y} \mid f)}{q_{Y|F}(\boldsymbol{y} \mid f)} \right) \\
&= P_0 \log \frac{P_0}{Q_0} + P_1 \log \frac{P_1}{Q_1} + \int_0^1 \mathcal{N}(y; z_P, \sigma_P^2) \log \frac{\mathcal{N}(y; z_P, \sigma_P^2)}{\mathcal{N}(y; z_Q, \sigma_Q^2)},
\end{aligned}
\tag{35}
$$

where we have $P_0 = \frac{1-\mathrm{erf}(z_P/(\sqrt{2}\sigma_P))}{2}$, $P_1 = \frac{1+\mathrm{erf}((z_P-1)/(\sqrt{2}\sigma_P))}{2}$, $Q_0 = \frac{1-\mathrm{erf}(z_Q/(\sqrt{2}\sigma_Q))}{2}$, $Q_1 = \frac{1+\mathrm{erf}((z_Q-1)/(\sqrt{2}\sigma_Q))}{2}$, and

$$\int_0^1 \mathcal{N}(y; z_P, \sigma^2) \log \frac{\mathcal{N}(y; z_P, \sigma_P^2)}{\mathcal{N}(y; z_Q, \sigma_Q^2)} dy =$$

$$= \int_0^1 \mathcal{N}(y; z_P, \sigma_P^2) \left( -\log \frac{\sigma_P}{\sigma_Q} - \frac{(y-z_P)^2}{2\sigma_P^2} + \frac{(y-z_Q)^2}{2\sigma_Q^2} \right) dy$$

$$= -(1 - P_0 - P_1) \log \frac{\sigma_P}{\sigma_Q} - \frac{\sigma_P}{2} \int_{\frac{-z_P}{\sigma_P}}^{\frac{1-z_P}{\sigma_P}} \mathcal{N}(t; 0, 1) t^2 dt + \frac{\sigma_Q}{2} \int_{\frac{-z_Q}{\sigma_Q}}^{\frac{1-z_Q}{\sigma_Q}} \mathcal{N}(t; 0, 1) t^2 dt$$

$$= -(1 - P_0 - P_1) \log \frac{\sigma_P}{\sigma_Q}$$

$$- \frac{\sigma_P}{2} \left( \frac{\mathrm{erf}\left(\frac{1-z_P}{\sqrt{2\sigma_P^2}}\right) - \mathrm{erf}\left(-\frac{z_P}{\sqrt{2\sigma_P^2}}\right)}{2} - \frac{1-z_P}{\sigma_P} \mathcal{N}\left(\frac{1-z_P}{\sigma_P}; 0, 1\right) - \frac{z_P}{\sigma_P} \mathcal{N}\left(-\frac{z_P}{\sigma_P}; 0, 1\right) \right)$$

$$+ \frac{\sigma_Q}{2} \left( \frac{\mathrm{erf}\left(\frac{1-z_Q}{\sqrt{2\sigma_Q^2}}\right) - \mathrm{erf}\left(-\frac{z_Q}{\sqrt{2\sigma_Q^2}}\right)}{2} - \frac{1-z_Q}{\sigma_Q} \mathcal{N}\left(\frac{1-z_Q}{\sigma_Q}; 0, 1\right) - \frac{z_Q}{\sigma_Q} \mathcal{N}\left(-\frac{z_Q}{\sigma_Q}; 0, 1\right) \right).$$

$$\tag{36}$$

## F   MAXIMUM DIRECT SUM ENTROPY OF MIXED DISTRIBUTIONS

Start by noting that the differential entropy of a Dirichlet random variable $Y \sim \mathrm{Dir}(\boldsymbol{\alpha})$ is

$$H(Y) = \log B(\boldsymbol{\alpha}) + (\alpha_0 - K)\psi(\alpha_0) - \sum_{k=1}^K (\alpha_k - 1)\psi(\alpha_k), \tag{37}$$

where $\alpha_0 = \sum_k^K \alpha_k$ and $\psi$ is the digamma function. When $\boldsymbol{\alpha} = \mathbf{1}$, this becomes a flat (uniform) density and the entropy attains its maximum value:

$$H(Y) = \log B(\boldsymbol{\alpha}) = -\log(K-1)!. \tag{38}$$

This value is negative for $K > 2$; it follows that the differential entropy of any distribution in the simplex is negative. We next determine the distribution $p_Y^{\oplus}(\boldsymbol{y})$ with the largest direct sum entropy. Considering only the maximal face, which corresponds to $\mathrm{ri}(\triangle_{K-1})$, the distribution with the largest entropy is the flat distribution, whose entropy is given in (38). In our definition of entropy in (10) this corresponds to a deterministic $F$ which puts all probability mass in this maximal face. At the opposite extreme, if we only assign probability to pure vertices, *i.e.*, if we constrain to *minimal* faces, a uniform choice leads to a (Shannon) entropy of $\log K$. We will show that looking at *all* faces further increases entropy.

Looking at (10), we see that the differential entropy term $H(Y \mid F = f)$ can be maximized separately for each $f$, the solution being the flat distribution on face $f \simeq \triangle_{k-1}$, which has entropy $-\log(k-1)!$, where $1 \le k \le K$. By symmetry, all faces of the same dimension $k-1$ look the same, and there are $\binom{K}{k}$ of them. Therefore the maximal entropy distribution is attained with $P_F$ of the form $P_F(f) = g(k)/\binom{K}{k}$ where $g : [K] \to \mathbb{R}_+$ is a function satisfying $\sum_{k=1}^K g(k) = 1$ (which can be regarded as a categorical probability mass function). If we choose a precision of $N$ bits, this

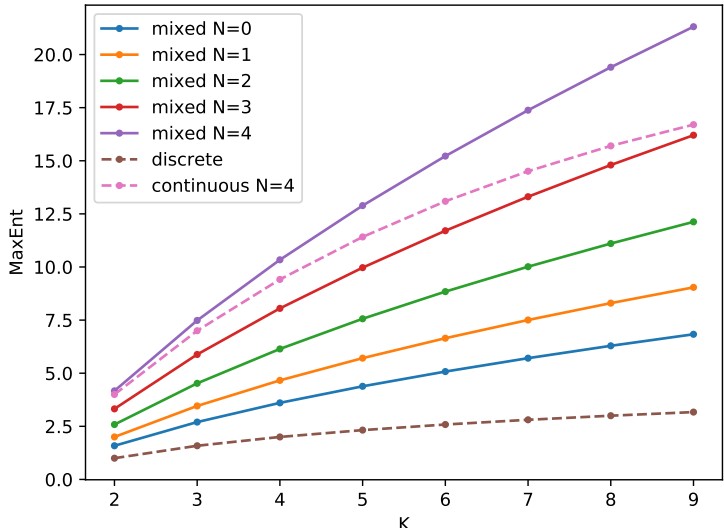

Figure 4: Maximum entropies for mixed distributions for several values of bit precision $N$, as a function of the simplex dimensionality $K - 1$. Shown are also the maximum entropies for the corresponding discrete and continuous cases, for comparison.

leads to:

$$
H_N^\oplus(Y) = - \sum_{f \in \bar{\mathcal{F}}(\triangle_{K-1})} P_F(f) \log P_F(f) + \sum_{f \in \bar{\mathcal{F}}(\triangle_{K-1})} P_F(f) \left( - \int_f p_{Y|F}(\boldsymbol{y} \mid f) \log p_{Y|F}(\boldsymbol{y} \mid f) \right)
$$
$$
+ N \sum_{k=1}^{K} (k-1) g(k) \tag{39}
$$
$$
= - \sum_{k=1}^{K} g(k) \log \frac{g(k)}{\binom{K}{k}} - \sum_{k=1}^{K} g(k)(\log(k-1)! - N(k-1))
$$
$$
= - \sum_{k=1}^{K} g(k) \log g(k) + \sum_{k=1}^{K} g(k) \log \frac{\binom{K}{k} 2^{N(k-1)}}{(k-1)!}. \tag{40}
$$

Note that the first term is the entropy of $g(\cdot)$ and the second term is a linear function of $g(\cdot)$, that is, (40) is a entropy-regularized argmax problem, hence the $g(\cdot)$ that maximizes this objective is the softmax transformation of the vector with components $\log \frac{\binom{K}{k} 2^{N(k-1)}}{(k-1)!}$, that is:

$$
g(k) = \frac{\frac{\binom{K}{k} 2^{N(k-1)}}{(k-1)!}}{\sum_{j=1}^{K} \frac{\binom{K}{j} 2^{N(j-1)}}{(j-1)!}}, \tag{41}
$$

and the maximum entropy value is

$$
H_{N,\max}^\oplus(Y) = \log \sum_{k=1}^{K} \frac{\binom{K}{k} 2^{N(k-1)}}{(k-1)!} = \log L_{K-1}^{(1)}(-2^N), \tag{42}
$$

where $L_n^{(\alpha)}(x)$ denotes the generalized Laguerre polynomial (Sonine, 1880), as stated in Proposition 2.

Figure 4 shows how the largest direct sum entropy varies with $K$, compared to the discrete and continuous entropies. For example, for $K = 2$, we obtain $g(1) = \frac{2}{2+2^N}$, $g(2) = \frac{2^N}{2+2^N}$, and $H_N^\oplus(Y) = \log(2 + 2^N)$, therefore, in the worst case, we need at most $\log_2(2 + 2^N)$ bits to encode

$Y \in \triangle_1$ with bit precision $N$. This is intuitive: the faces of $\triangle_1$ are the two vertices $\{(0,1)\}$ and $\{(1,0)\}$ and the line segment $[0,1]$. The first two faces have a probability of $\frac{1}{2+2^N}$ and the last one have a probability $\frac{2^N}{2+2^N}$. To encode a point in the simplex we first need to indicate which of these three faces it belongs to (which requires $\frac{2}{2+2^N}\log_2(2+2^N) + \frac{2^N}{2+2^N}\log_2\frac{2+2^N}{2^N} = \log_2(2+2^N) - \frac{N2^N}{2+2^N}$ bits), and with $\frac{2^N}{2+2^N}$ probability we need to encode a point uniformly distributed in the segment $[0,1]$ with $N$ bit precision, which requires extra $\frac{N2^N}{2+2^N}$ bits on average. Putting this all together, the total number of bits is $\log_2(2+2^N)$, as expected.

## G EXPERIMENTAL DETAILS

In this section, we give further details on the architectures and hyperparameters used in our main experiments (App. G.1, App. G.2, and App.G.3), which are focused on learning sparse representations, in App. G.4 we experiment with the Mixed Dirichlet as a likelihood function to model simplex-valued data. Finally, in App G.5, we describe our computing infrastructure.

### G.1 EMERGENT COMMUNICATION GAME

Let $\mathcal{V} = \{v_1, \ldots, v_{|\mathcal{V}|}\}$ be the collection of images from which the sender sees a single image $v_j$. We consider a latent variable model with observed variables $x = (\mathcal{V}, j) \in \mathcal{X}$ and latent stochastic variables $y$ chosen from a vocabulary set $\mathcal{Y}$. We let the sender correspond to the probability model $\pi(y|x,\theta) = p(y \mid v_j, \theta_\pi)$ of the latent variable and the receiver be modeled as $p(j \mid \mathcal{V}, y, \theta_\ell)$. The overall fit to the dataset $\mathcal{D}$ is $\sum_{x \in \mathcal{D}} \mathcal{L}_x(\theta)$, where we marginalize the latent variable $y$ to compute the loss of each observation, that is

$$\mathcal{L}_x(\theta) = \mathbb{E}_{\pi(y|x,\theta)}\left[\ell(x,y;\theta)\right] = \sum_{y \in \mathcal{Y}} \pi(y|x,\theta)\,\ell(x,y;\theta), \tag{43}$$

where the receiver is used to define the downstream loss, $\ell(x,y;\theta) := -\log p(j \mid \mathcal{V}, y, \theta_\ell)$. Notably, we do not add a (discrete) entropy regularization term of $\pi(y \mid x, \theta)$ to (43) with a coefficient as an hyperparameter as in Correia et al. (2020).

**Data.**  The dataset consists of a subset of ImageNet (Deng et al., 2009)[5] containing 463,000 images that are then passed through a pretrained VGG (Simonyan & Zisserman, 2015), from which the representations at the second-to-last fully connected layers are saved and used as input to the sender and the receiver. To get the dataset visit `https://github.com/DianeBouchacourt/SignalingGame` (Bouchacourt & Baroni, 2018).

**Architecture and hyperparameters.**  We follow the experimental procedure described in Correia et al. (2020): the architecture of both the *sender* and the *receiver* are identical to theirs, we set the size of the collection of images $|\mathcal{V}|$ to 16, the size of the vocabulary of the sender to 256, the hidden size to 512, and the embedding size to 256. We choose the best hyperparameter configuration by doing a grid search on the learning rate (0.01, 0.005, 0.001) and, for each configuration, evaluating the communication success on the validation set. For the Gumbel-Softmax models, the temperature is annealed using the schedule $\tau = \max(0.5, \exp -rt)$, where $r = 1e-5$ and $t$ is updated every $N = 1000$ steps. For the $K$-D Hard Concrete we use a scaling constant $\lambda = 1.1$ and for Gaussian-Sparsemax we set $\Sigma = I$. All models were trained for 500 epochs using the Adam optimizer with a batch size of 64.

### G.2 BIT-VECTOR VAE

In this experiment, the training objective is to minimize the negative ELBO: we rewrite (43) with $\ell(x,y;\theta_\ell) = -\log\frac{p(x,y|\phi)}{q(y|x,\lambda)}$, where we decompose the approximate posterior as $q(y \mid x, \lambda) = \prod_{i=1}^{D} q(y_i \mid x, \lambda)$, with $D = 128$ being the number of binary latent variables $y_i$. For each bit, we

---

[5]Available for research under the terms and conditions described in `https://www.image-net.org/download`.

consider a uniform prior $p(y)$ when the approximate posterior is purely discrete or continuous, and the maxent mixed distribution (Prop. 2) for the mixed cases.

**Data.**    We use Fashion-MNIST (Xiao et al., 2017)[6], comprising of $28 \times 28$ grayscale images of fashion products from different categories.

**Architecture and hyperparameters.**    We follow the architecture and experimental procedure described in Correia et al. (2020), where the inference and generative network consist of one 128-node hidden layer with ReLU activation functions. We choose the best hyperparameter configuration by doing a grid search on the learning rate (0.0005, 0.001, 0.002) and choosing the best model based on the value of the negative ELBO on the validation set. For the Gumbel-Softmax models, the temperature is annealed using the schedule $\tau = \max(0.5, \exp -rt)$, where we search $r \in \{1e-5, 1e-4\}$ and $t$ is updated every $N = 1000$ steps; also, for these models, we relax the sample into the continuous space but assume a discrete distribution when computing the entropy of $\pi(y \mid x, \theta)$, leading to an incoherent evaluation for Gumbel-Softmax without ST. For the Hard Concrete distribution we follow Louizos et al. (2018) and stretch the concrete distribution to the $(-0.1, 1.1)$ interval and then apply a hard-sigmoid on its random samples. For the Gaussian-Sparsemax, we use $\sigma^2 = 1$. All models were trained for 100 epochs using the Adam optimizer with a batch size of 64.

## G.3    Mixed-Latent VAE on MNIST

**Data.**    We use stochastically binarized MNIST (LeCun et al., 2010).[7] The first 55,000 instances are used for training, the next 5,000 instances for development and the remaining 10,000 for test.

**Architecture and hyperparameters.**    The model is a VAE with a $K$-dimensional latent code (with $K = 10$ throughout), both the decoder (which parametrizes the observation model) and the encoder (which parametrizes the inference model) are based on feed-forward neural networks. The decoder maps from a sampled latent code $z$ to a collection of $D = 28 \times 28$ Bernoulli distributions. We use a feed-forward decoder with two ReLU-activated hidden layers, each with 500 units. The encoder maps from a $D$-dimensional data point $x$ to the parameters of a variational approximation to the model's posterior distribution. For Gaussian models we predict $K$ locations and $K$ scales (with softplus activation). For Dirichlet models we predict $K$ concentrations using softplus clamped to $[10^{-3}, 10^3]$. For Mixed Dirichlet models we predict $K$ log-potentials clamped to $[-10, 10]$, and $K$ concentrations using softplus clamped to $[10^{-3}, 10^3]$. For regularization we employ dropout and L2 penalty. We use two Adam optimizers, one for the parameters of the decoder and another for the parameters of the encoder, each with its own learning rate. We search for the best configuration of hyperparameters using importance-sampled estimates of negative log-likelihood of the model given a development set. The parameters we consider are: learning rates (in $\{10^{-5}, 10^{-4}, 10^{-3}\}$, and their halves), regularization strength (in $\{0, 10^{-7}, 10^{-6}, 10^{-5}\}$), and dropout rate (in $\{0, 0.1, 0.2, 0.3\}$). All models are trained for 500 epochs with stochastic mini batches of size 100. During training, for gradient estimation, we sample each latent variable exactly once. For importance sampling, we use 100 samples for model selection, and 1000 samples for evaluation on the test set.

**Gradient estimation.**    We use single-sample reparametrized gradient estimates for most models: Gaussian (unbiased), Dirichlet (unbiased), Gumbel-Softmax ST (biased due to straight-trough), and Mixed Dirichlet (unbiased). A gradient estimate for Mixed Dirichlet models, in particular, has two sources of stochasticity, one from the implicit reparametrization for Dirichlet samples, another from score function estimation for Gibbs samples. To reduce variance due to score function estimation, the Mixed Dirichlet model we report employs an additional sample, used to compute a self-critic baseline. We have also looked into a simpler variant, which dispenses with this self-critic baseline, and instead employs a simple running average baseline, this model performed very close to that with a self-critic. The purely discrete latent variable model (Categorical) uses the exact gradient of the ELBO, given a mini batch, which we obtain by exactly marginalizing the $K = 10$ assignments of the latent variable.

---

[6]License available at `https://github.com/zalandoresearch/fashion-mnist/blob/master/LICENSE`.

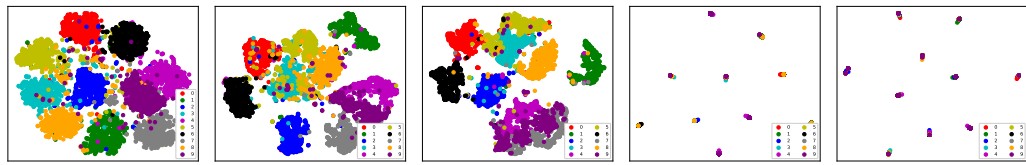

Figure 5: tSNE plots of posterior samples, from left-to-right: Gaussian, Dirichlet, Mixed Dirichlet, Categorical, Gumbel-Softmax ST.

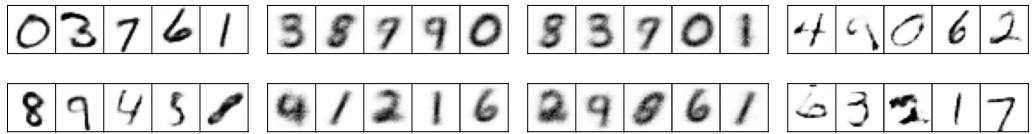

Figure 6: Pixel-wise average of 100 model samples generated by conditioning on each of the ten vertices of the simplex. From left-to-right: Mixed Dirichlet, Categorical, Gumbel-Softmax ST, and Dirichlet (as the Dirichlet does not support the vertices of the simplex, we add uniform noise (between 0 and 0.1) to each coordinate of a vertex and renormalize).

**Prior, posterior, and KL term in the ELBO.** The Gaussian model employs a standard Gaussian prior and a parametrized Gaussian posterior approximation, the KL term in the ELBO is computed in closed-form. The Dirichlet model employs a uniform Dirichlet prior (concentration $1$) and a parametrized Dirichlet posterior approximation, the KL term in the ELBO is also computed in closed-form. Both the Categorical and the Gumbel-Softmax ST models employ a uniform Categorical prior and a Categorical approximate posterior (albeit parametrized via Gumbel-Softmax ST for the latter model), the KL term in the ELBO is between two Categorical distributions and thus computed in closed form. The Mixed Dirichlet model employs a maximum entropy prior with bit-precision parameter $N = 0$ and a parametrized Mixed Dirichlet approximate posterior, the KL term in the ELBO is partly exact and partly estimated, in particular, the contribution of the Gibbs distributions over faces is computed in closed form.

**Additional plots.** Figure 5 shows tSNE plots where each validation digit is encoded by a sample from the approximate posterior obtained by conditioning on that digit. The continuous (Gaussian and Dirichlet) and mixed (Mixed Dirichlet) models learn to represent the digit rather well, with some plausible confusions by the Mixed Dirichlet model (*e.g.*, 4, 7, and 9). The models that can only encode data using the vertices of the simplex (Categorical and Gumbel-Softmax ST) typically map multiple digits to the same vertex of the simplex. For models whose priors support the vertices of the simplex (Mixed Dirichlet, Categorical, and Gumbel-Softmax ST), we can inspect what the vertices of the simplex typically map to (in data space). Figure 6 shows 100 such samples per vertex. We also include samples from the Dirichlet model, though note that the Dirichlet prior does not support the vertices of the simplex, thus we sample from points very close to the vertices (but in the relative interior of the simplex). We also report conditional and unconditional generations from each model in Figures 7 and 8.

### G.4 SIMPLEX-VALUED REGRESSION

Simplex-valued data are observations in the form of probability vectors, they appear in statistics in contexts such as time series (*e.g.*, modeling polling data) and in machine learning in contexts such as knowledge distillation (*e.g.*, teaching a compact network to predict the categorical distributions predicted by a larger network).

[7]Available under the terms of the Creative Commons Attribution-Share Alike 3.0 license (`https://creativecommons.org/licenses/by-sa/3.0/`).

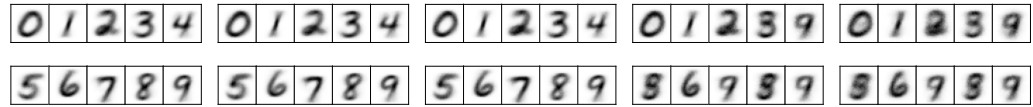

Figure 7: Conditional generation. For each instance of each class in the validation set (note that none of the models has access to the label), we sample a latent code conditioned on the digit, and re-sample a digit from the model. The illustration displays the pixel-wise average across all validation instances of the same class. From left-to-right: Gaussian, Dirichlet, Mixed Dirichlet, Categorical, Gumbel-Softmax ST.

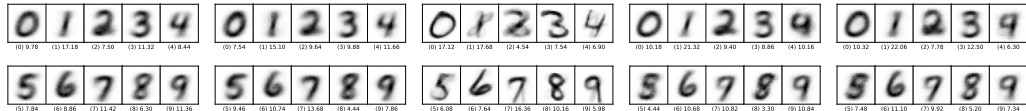

Figure 8: Unconditional generation. For each model, we sample a digit via ancestral sampling (*i.e.*, $z$ is sampled from the prior, then, given $z$, $x$ is sampled from the observation model). We gather 5000 such samples and group them by class as predicted by a 5-nearest neighbour classifier trained on the MNIST training set (we use `kd tree` from scikit-learn), the classifier achieves 95% F1 on the MNIST test set. Each image in the illustration is a pixel-wise average of the samples in the cluster, we also report the percentage of digits in each cluster. From left-to-right: Gaussian, Dirichlet, Mixed Dirichlet, Categorical, Gumbel-Softmax ST.

**Data and task.** We experiment with the UK election data setup by Gordon-Rodriguez et al. (2020, Section 5.2).[8] The UK electorate is partitioned into 650 constituencies, each electing one member of parliament in a winner-takes-all vote. Hence the data are 650 observed vectors of proportions over the four major parties plus a 'remainder' category (*i.e.*, each observation is a point in the $\Delta_{5-1}$ simplex, including its faces). Modeling simplex-valued data with the Dirichlet distribution is tricky for the Dirichlet does not support sparse outcomes. While pre-processing the data into the relative interior of the simplex is a simple strategy (*e.g.*, add small positive noise to coordinates and renormalize), it is ineffective for the Dirichlet pdf either diverges or vanishes at extrema (neighbourhoods of the lower-dimensional faces of the simplex). Gordon-Rodriguez et al. (2020) document this and other difficulties in modeling with the Dirichlet likelihood function. To address these limitations they develop the *continuous categorical* (CC) distribution, an exponential family that supports the entire simplex (*i.e.*, it assigns non-zero density to any point in the simplex), and does not diverge at the extrema. The CC enjoys various analytical properties, but it still cannot assign non-zero mass to the lower-dimensional faces of the simplex, thus while it is a better choice of likelihood function than the Dirichlet, CC samples are never truly sparse (thus test-time predictions are always dense).

**Architecture and hyperparameters.** For Dirichlet and CC, we use a linear layer to map from the input predictors to 5 log-concentration parameters. For Mixed Dirichlet we use 2 linear layers: one maps from the input predictors to 5 scores (clamped to $[-10, 10]$) which parametrize $P_F$, the other maps to 5 strictly positive concentrations (we use softplus activations, with pre activations constrained to $[-10, 10]$) which parametrize $P_{Y|F=f}$. We train all models using Adam with learning rate 0.1 and no weight decay for exactly 400 steps without mini-batching with 20% of the available data used for training. Following Gordon-Rodriguez et al. (2020), we pre-process the data (add $10^{-3}$ to each coordinate and renormalize) for Dirichlet and CC, but this is not done for Mixed Dirichlet.

**Results.** Figure 9 compares the three choices of likelihood function. We report two prediction rules for our Mixed Dirichlet model. *Sample mean*: we predict stochastically by drawing 100 samples and outputting the sample mean. *Most probable mean*: we predict deterministically by outputting the mean of the Dirichlet on the face that is assigned highest probability by the model (finding this *most probable face* is an operation that takes time $\mathcal{O}(K)$, and recall $K = 5$ in this task). We can see that Mixed Dirichlet does not suffer from the pathologies of the Dirichlet, due to face stratification, and lowers test error a bit more than CC (see Table 9), likely due to the ability to sample actual zeros.

---

[8]https://commonslibrary.parliament.uk/research-briefings/cbp-8749/

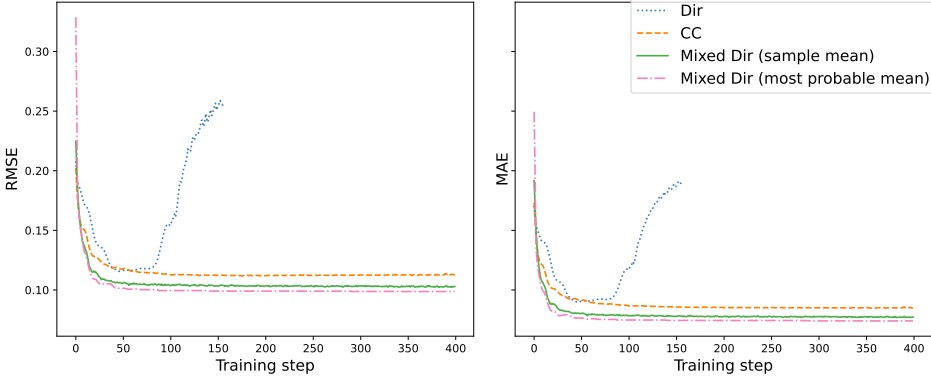

Figure 9: Test error (root-mean-square error on the left, mean absolute error on the right) of generalized linear regression towards 5-dimensional vote proportions (UK election data). We compare 3 likelihood functions: Dirichlet, Continuous Categorical (CC), and Mixed Dirichlet.

The Mixed Dirichlet uses twice more parameters (we need to parametrize two components), but training time is barely affected (sampling and density assessments are all linear in $K$), the training loss and its gradients are stable, and the algorithm converges just as early as CC's. As the Mixed Dirichlet produces sparse samples, it is interesting to inspect how often it succeeds to predict whether an output coordinate is zero or not (*i.e.*, whether $y_k > 0$, which is true for 77.3% of the targets in the text set). The sample mean predicts whether $y_k > 0$ with macro F1 0.92, whereas the most probable mean achieves macro F1 0.94.

| Model | RMSE | MAE |
|---|---|---|
| CC | 0.1124 | 0.0847 |
| Mixed Dirichlet | | |
|   sample mean | 0.1030 | 0.0774 |
|   most probable mean | 0.0987 | 0.0740 |

Table 4: Test root-mean-square error and mean absolute error as a function of choice of likelihood function and prediction rule.

### G.5 COMPUTING INFRASTRUCTURE

Our infrastructure consists of 5 machines with the specifications shown in Table 5. The machines were used interchangeably, and all experiments were executed in a single GPU. Despite having machines with different specifications, we did not observe large differences in the execution time of our models across different machines.

Table 5: Computing infrastructure.

| # | GPU | CPU |
|---|---|---|
| 1. | $4 \times$ Titan Xp - 12GB | $16 \times$ AMD Ryzen 1950X @ 3.40GHz - 128GB |
| 2. | $4 \times$ GTX 1080 Ti - 12GB | $8 \times$ Intel i7-9800X @ 3.80GHz - 128GB |
| 3. | $3 \times$ RTX 2080 Ti - 12GB | $12 \times$ AMD Ryzen 2920X @ 3.50GHz - 128GB |
| 4. | $3 \times$ RTX 2080 Ti - 12GB | $12 \times$ AMD Ryzen 2920X @ 3.50GHz - 128GB |
| 5. | $2 \times$ GTX Titan X - 12GB | $12 \times$ Intel Xeon E5-1650 v3 @ 3.50GHz - 64 GB |

