# OpenReview forum: "Sparse Communication via Mixed Distributions"
_ICLR.cc/2022/Conference — ICLR 2022 Oral_

### Official Review · Reviewer_ASbt · 2021-11-02

**Correctness:** 2
**Technical Novelty And Significance:** 3
**Empirical Novelty And Significance:** 3
**Recommendation:** 6
**Confidence:** 2

**Main Review:**

The authors are encouraged for their good work.
Though they do try to introduce "mixed variables" (ie random variables consisted by both categorical and continuous data) there are a few unclear points: for example, they consider a (ie one) categorical variable with alphabet K. What is the case when there are more than one categorical points, with different alphabet (eg ordinal data with alphabet X) ?
Also, what is the algorithmic complexity with respect to the number of observations, the number/rate of continuous and discrete outcomes and the number of variables (in the case of multivariate mixed variable?
Finally, Table 1 is quite unclear to me.

**Summary Of The Paper:**

The authors wish to introduce a new kind of random variables that are consisted of both continuous and categorical data.
They also provide the corresponding theory and take into account information theoretical aspects.

**Summary Of The Review:**

My main point is that the authors
1) do not provide information regarding the case of having more than one discrete outcomes with different alphabets;
2) what is the algorithmic complexity with respect to the number of observations, the rate of continuous vs discrete outcomes;
3) they still work on the code and they do not provide it unless the paper is accepted

---

> ### Author Response · Authors · 2021-11-22
> **Author response**
>
> Thank you for the review! We will now try to address your concerns about our paper.
>
> > Table 1 is quite unclear to me.
>
> Table 1 is a summary table comparing the properties of the discrete, continuous, and mixed distributions considered in our paper. The table distinguishes distributions according to three questions:
>
> 1. Does the distribution assign probability mass to all the faces of the simplex? (for visualization, see Figure 1, which highlights two 3D distributions with opposite answers);
> 2. Is it restricted to binary variables or does it support multiple categories (K >= 2)?;
> 3. For mixed distributions, is it characterized extrinsically (using the “sample-and-project” approach discussed in Section 3.3) or intrinsically (based on face stratification; a mixture of distributions is specified directly over the faces of the probability simplex)?
>
> Please let us know if this clarifies your question.
>
> > do not provide information regarding the case of having more than one discrete outcomes with different alphabets;
>
> Do you mean predicting a tuple of T discrete outcomes instead of a single one? Our method handles this case as other methods would, namely, we can give the T variables independent treatment, or factorize their joint probability via the chain rule, in which case each factor is a conditional over a single mixed variable. Our techniques apply to both cases without special treatment.
>
> If the T variables are treated independently, this corresponds to a latent space which is a product of simplices of (possibly) different dimensions -- in this case the total direct sum entropy/KL is the sum of entropies/KL as expected. The binary vector case that we address in our paper for the bit-vector VAE (Section 5) is a particular case of a product of simplices with |V|=2. You can check Section 3.1: “the hypercube can be regarded as a product of simplices...”
>
> If the T variables are not treated independently, we can have multiple random variables and use a joint distribution via the chain rule $p(x_1, ..., x_N) = prod_{i=1}^N p(x_i|x_{<i})$, where the distribution of each $x_i$ is a mixed distribution parameterized given $x_{<i}$.
>
> In summary, there is nothing specific to our mixed distributions that would provide any limitation when extending to multiple discrete outcomes. Does this answer your question?
>
> > what is the algorithmic complexity with respect to the number of observations, the rate of continuous vs discrete outcomes;
>
> Do you mean the algorithm complexity of sampling a face? This is relevant for our proposed intrinsic mixed distribution, the Mixed Dirichlet. In that case, sampling is done with the dynamic program described in App. C (to rule out the empty face), and the complexity is O(K). You can check our App. C for details about the computation complexity of other quantities of interest. For instance, both the log-normalizer and the gradient of KL(P_F||Q_F) can be computed in a single pass through a DAG of size O(K); the exact computation of the continuous part in H(Y|F) costs O(2^K) but, for large K, we can use an MC estimate that costs O(K).
>
> For the multivariate Gaussian-Sparsemax, the complexity is that of sampling K i.i.d. numbers from a Gaussian and projecting them to the simplex (which is the asymptotic cost of sparsemax, O(K), or O(K log K) with a sorting algorithm [1]).
>
> The algorithmic complexity is linear in the number of observations, for a fixed number of epochs.
>
> Related to your previous question, this is what happens if we had a tuple of T mixed random variables. With independent mixed random variables, the sampling is independent, thus the time complexity is unchanged on GPUs, the space grows linearly with the number of random variables. With a chain rule factorization, the complexity is linear because we sample the $i$-th random variable given an assignment of the preceding ones.
>
> [1] André FT Martins and Ramón Fernandez Astudillo. From softmax to sparsemax: A sparse model of attention and multi-label classification. In ICML, 2016.
>
> > they still work on the code and they do not provide it unless the paper is accepted
>
> Please note that we included all the code and instructions to reproduce our experimental results as supplementary material (please check the submitted zip file). This is described in our reproducibility statement (see page 10). The code is finalized and it will be publicly released on github after the anonymity period ends. We hope this clarifies your concern.

---

> > ### Comment · Reviewer_ASbt · 2021-11-23
> > **Response to authors**
> >
> > Dear Authors,
> >
> > Many thanks for your detailed response
> >
> > My two major concerns regarding:
> > 1) how would you treat more than one tuples of T discrete outcomes, and ;
> >
> > 2) the algorithmic complexity;
> >
> > where clearly explained by your response.
> >
> > On (2) I was mainly worried about the the exact computation of the continuous part which costs as you said O(2^K) but, you responded that for large K, you can by-pass the exponential problem by using an MC estimate that costs O(K) (it is important though to acknowledge that this is merely an estimate).
> >
> > I thank the authors for their detailed response; my evaluation will be changed accordingly
> >
> > Good job.

---

### Official Review · Reviewer_F3KA · 2021-11-03

**Correctness:** 4
**Technical Novelty And Significance:** 3
**Empirical Novelty And Significance:** 3
**Recommendation:** 8
**Confidence:** 3

**Main Review:**

**Reasons to accept:**
- The paper is well-written and appears relevant to recent developments.
- The motivation of the paper in relevance to bridging the continuous representations computed by neural networks and other machine learning models with the discrete representations that characterize humans is interesting. Table 1 nicely summarizes the contributions of this work.
- The mathematical coverage also seems reasonable.
- Extensive experiments on three different tasks, including an emergent communication benchmark. Supplementary material also provides qualitative examples.

**Suggested improvements:**
- Mixed random variables are a standard topic in probability theory. Definitions 1,2,3 appear to be natural in this theoretical context and the same applies to Propositions 1,2. More references in the mathematical and probability literature would be helpful due to the coverage of mixed random variables and associated theory in this literature.

**Update after rebuttal:** I have read the reviews and the author's responses. This is solid well-written work, my review remains unchanged.

**Summary Of The Paper:**

This paper presents multidimensional extensions for mixed random variables originating from discrete-continuous hybrids based on truncation and rectification, which have been proposed for univariate distributions. The proposed extension replaces truncation by sparse projections to the simplex. The authors also propose a direct sum base measure definition on the face lattice of the probability simplex and intrinsic sampling strategies motivated by “manifold stratification”. Based on these introductions, new entropy and Kullback-Leibler divergence functions that subsume the discrete and differential cases and have interpretations in terms of code optimality, are presented.

**Summary Of The Review:**

This is a paper with mathematical rigor and connections to emergent communication, which is a current topic of interest for specific machine learning communities. Albeit the submission of this paper would better fit a probability and statistics venue, to the best of my knowledge, the definitions 1-3 and propositions 1,2 appear to be novel.

---

> ### Author Response · Authors · 2021-11-22
> **Author response**
>
> Thank you for your positive review and suggestions! Our paper currently mentions some mathematical and probability literature that can be helpful to the reader, including Ziegler (1995), Conway (2019), and we added a new  reference to Halmos’ “Measure Theory” book in the revised version. Any further recommendations are very welcome!

---

### Official Review · Reviewer_ePyA · 2021-11-03

**Correctness:** 4
**Technical Novelty And Significance:** 4
**Empirical Novelty And Significance:** 3
**Recommendation:** 8
**Confidence:** 3

**Main Review:**

Strengths:

1. The technique in this paper is very solid. I believe the theoretical foundations built in this paper will be useful in the future research on mixed random variables.

2. The idea of this paper makes sense, and this paper is well written and easy to read.

3. The experimental results demonstrate the usefulness of the proposed framework.


Weaknesses:
1. My only concern is that whether ICLR is a proper conference to publish this paper, since it involves some concepts about measure theory and probability theory.


**Summary Of The Paper:**

In this paper, the authors build rigorous theoretical foundations for mixed random variables. They first define a natural measure for mixed random variables, which looks like a direct sum. Then they define the entropy and KL divergence based on the proposed measure. They also give two strategies for representing and sampling mixed random variables. At last, they conduct some experiments to illustrate the usefulness of their framework.

**Summary Of The Review:**

This paper is very solid. It builds some important theoretical foundations for the mixed random variables. It seems that this will be useful for the future research on this topic.

---

> ### Author Response · Authors · 2021-11-22
> **Author response**
>
> Thank you for your positive review!
>
> While our paper has indeed a strong theoretical component, we do not see this as a weakness. Our paper addresses the problem of learning sparse representations, a topic of wide interest to the ICLR audience, by carefully building a sound mathematical framework for handling mixed distributions. Other ICLR papers which ally theory and practice for learning sparse representations include the paper which proposed the Concrete distribution (ICLR 17, https://arxiv.org/pdf/1611.00712.pdf), D-VAE (ICLR17, https://arxiv.org/pdf/1609.02200.pdf), and Hard Concrete (ICLR 18, https://arxiv.org/pdf/1712.01312.pdf), among several others, all of which a strong inspiration to our own work. Alongside our theoretical contributions, we demonstrate with a set of simple experiments that our framework can be useful in practice, with mixed distributions being a promising way to address some of the limitations of latent variable models. Having said that, we welcome any suggestions that could make our paper more accessible to an audience less familiar with measure/probability theory.

---

### Official Review · Reviewer_B5yQ · 2021-11-07

**Correctness:** 4
**Technical Novelty And Significance:** 4
**Empirical Novelty And Significance:** 3
**Recommendation:** 8
**Confidence:** 5

**Main Review:**

This paper was a pleasure to read. The idea is simple and intuitive, and it addresses a recurring issue with commonly-used simplex-valued distributions, allowing to better model sparcity while avoiding diverging likelihoods in the presence of 0s. The paper is clearly written, and the mathematical exposition is formal and well presented. I also believe the idea presented in this paper will be easily used to propose new simplex-valued distributions and be valuable to the community: while the authors propose several instances of mixed distributions, one can easily think of potential alternatives.

My only complaint about the paper is that the experiments focus mostly on sparcity, and not on avoiding ill-defined log-likelihoods, which I actually believe is another benefit of the proposed distributions. For example, the distribution proposed in [1], which is discussed in the paper, addresses learning $p(y|x)$ when $y$ is simplex-valued in the presence of 0s in the data; which can be handled in a more principled way through mixed distributions. Finally, the experiments are carried out against the Gumbel-Softmax, but not more recent improvements upon it, e.g. [2,3,4]. Nevertheless, I believe the contributions of the paper are enough to warrant publication.

Minor things:

-Section 3.1, when defining a face, I believe "A face P is any intersection of P with a halfspace such that none of the interior points of P lie on the boundary of the halfspace" should be replaced by "A face P is any intersection of P with a closed halfspace such that none of the interior points of P lie on the boundary of the halfspace" for added clarity.


[1] The continuous categorial: A novel simplex-valued exponential family, Gordon-Rodriguez et al., ICML 2020

[2] Estimating Gradients for Discrete Random Variables by Sampling Without Replacement, Kool et al., ICLR 2020

[3] Invertible Gaussian Reparameterization: Revisiting the Gumbel-Softmax, Potapczyski et al., NeurIPS 2020

[4] Rao-Blackwellizing the Straight-Through Gumbel-Softmax Gradient Estimator, Paulus et al., ICLR 2021

========================================================================================================

UPDATE 1 AFTER REBUTTAL

========================================================================================================

I have read the author's rebuttal as well as the other reviews, and my opinion on the paper remains that it should be accepted. I particularly appreciate the authors adding the suggested additional experiment to an already strong paper.

**Summary Of The Paper:**

This paper proposes mixed distributions over convex polytopes such as the probability simplex. The proposed distributions are a discrete mixture over the faces of the polytope of "continuous" distributions on the corresponding face (formally, absolutely continuous wrt the Lebesgue measure on the face). For example, for the 2-simplex there is a distribution over the triangle, over each edge, and over each vertex. The authors formalize the dominating measure of these distributions, which they call the direct sum measure (given by the mixture over the counting measure over faces and Lebesgue measures on the faces); and derive formulas for entropy and KL divergences between such distributions, as well as characterizing maximum-entropy distributions.

**Summary Of The Review:**

This paper proposes a well-motivated and mathematically elegant family of distributions on convex polytopes, allowing to place positive probability not only on the interior of the polytope, but on its faces as well. While there is room for improving the experiments, I believe the presented experiments and theoretical developments -- which can easily enable future work -- should be accepted.

---

> ### Author Response · Authors · 2021-11-22
> **Author response**
>
> Thank you for your positive reviews and suggestions!
>
> In the manuscript we focused on learning sparse representations, but the direction you suggest is indeed very interesting, thanks for pointing it out. We have now reproduced one of the experiments reported by [1] comparing three likelihood functions in a generalized linear model: Dirichlet, the Continuous Categorical [1], and our Mixed Dirichlet. For each observation, the model regresses from 4 predictors to a 5-dimensional probability vector (proportions of votes towards one of 5 parties in an election), you can find the results in Appendix G.4. As you expected, the Mixed Dirichlet too addresses the pathologies of the Dirichlet in this setting, and it also shows a slight advantage over the Continuous Categorical, likely due to the fact that Mixed Dirichlet samples are often sparse.
>
> > Section 3.1, when defining a face, I believe (...) should be replaced by "A face P is any intersection of P with a closed halfspace such that none of the interior points of P lie on the boundary of the halfspace" for added clarity.
>
> Thank you for the suggestion! We have updated the manuscript.

---

### Decision · Program_Chairs · 2022-01-20

**Decision:**

Accept (Oral)

**Comment:**

This paper proposes mixed distributions over convex polytopes, and provides theory for mixed distributions that is relevant to the machine learning community. All of the reviewers were positive, and agree that this is a solid contribution. I agree, and I believe that this paper stands a chance of being a foundational paper for future work in probabilistic ML and structured learning.